# A Review of Wood Compression along the Grain—After the 100th Anniversary of Pleating

**Mátyás Báder ***[ID] **and Róbert Németh**

Institute of Wood Technology and Technical Sciences, Faculty of Wood Engineering and Creative Industries, University of Sopron, Bajcsy-Zsilinszky 4, 9400 Sopron, Hungary; nemeth.robert@uni-sopron.hu
* Correspondence: bader.matyas@uni-sopron.hu; Tel.: +36-30-997-4426

**Abstract:** This study focuses on the compression of wood along the grain (also known as pleating), a modification that improves the pliability of higher-density hardwoods with a moisture content above 20%. Pleated wood can be bent into small curves in any direction. The success of the industrial pleating process in some parts of the world is influenced by many factors, such as wood species, wood quality, moisture content, compression ratio, fixation time, etc. Pleating by 20% causes the modulus of elasticity to decrease to one-third for oak and beech, and the bending ratio can be increased above $\frac{1}{2}$. Bending stress decreases and the absorbed energy increases multiple times. The impact bending strength also increases significantly. The walls of cells crinkle by pleating and the microfibrils of the fibres become distorted. Many patents, articles, and books on this subject have been published since 1917, and this review attempts to introduce and, where necessary, critically analyse them.

**Keywords:** cell structure; fibre softening; hardwood; longitudinal compression; mechanical properties; physical properties; pleating; wood bending; wood modification





## 1. Introduction

Since the turn of the millennium, the topic of wood modification has become more prominent. Many publications on the subject have appeared, but compression along the grain is barely mentioned within mechanical wood modifications (Van Acker and Hill [1], Hill [2], Sandberg and Navi [3], Sandberg et al. [4], Kamke [5], and Sandberg et al. [6], among others). There is no extensive literature on the subject; therefore, this paper tries to fill the gap in the review of wood compression along the grain because although a lot of knowledge from the last century is available, it is not possible to find it in a single paper or summarized and processed in a transparent way. As concluded by Harvey in 1984: "No man is civilised or mentally adult until he realises that the past, the present, and the future are indivisible." This statement applies equally to all fields of science and technology [7], and this is the reason this study provides a detailed overview of the history of wood compression along the grain, its raw materials, the process of compression, the posttreatment opportunities, and the changes in physical and mechanical properties and cellular structure, using both industrial and scientific information. The authors have tried to carry out an in-depth review and apologize if a study or patent published over the past 100 years from any part of the world is not mentioned.

Nowadays, curved wood constituents are usually made of glued or glued-laminated solid wood or wood-based composite materials [8]. These technologies require a lot of wood, tools, glue, and precise work. It follows that the properties of wood must be modified so that it can be shaped by simple tools using little force. This is possible with wood compressed along its grain. As shown in Section 5.4, the cell walls of the wood become wavy as a result of this thermo-hydromechanical (THM) wood modification process, similarly to a plisse shade, which is why Báder and Németh [9] proposed the term "pleating" to describe the phenomenon and to briefly name the modification process. Pleating is basically the process

of compressing high-quality hardwood along the grain after plasticization by supporting the sides of the wood to avoid buckling. Many treatment variations can be used with a wide range of parameters, as will be shown later in Section 4, but the final result is always a timber free from harmful substances and highly bendable in the cold state.

Through pleating, it became possible to create pliable, springy, and light furniture made of bent wood [10]. Pleated wood bent to the proper shape can be mechanically fixed as a handrail or fixed by glue as an edge closure without cracking or breaking of the material [11,12]. Veneer can also be made of it, which can be used for 3D curved surfaces where some elongation of the veneer is required [10,13]. The shape of wooden spiral springs corresponds to those used in mechanical engineering. At the same size, the energy stored in a wooden spring is smaller compared to a metal spring, but it can be used in many areas of life as a bio-product due to its low weight and the fact that it is non-magnetizable [14]. Segesdy [15] investigated the usability of pleated wood for kitchen furniture. He found that it primarily contributes to the aesthetic functions of the kitchen, such as fittings, decorative and design accessories, and supplementary devices. It can also be used for other furniture in the apartment, e.g., chair legs, armrests, backrests, and table legs [16]. Sőregi [17] used pleated wood to design the interior of a boat cabin, which is ideal to produce unique curved equipment as safe furniture for ships and aircraft, or to produce reinforcing ribs [18]. Kollmann [16] also described its use as a raw material for certain parts of a wide variety of vehicles. Because it can be individually manufactured, it is also very useful in restoration, where it is not necessary to rebuild the contemporary technology to reproduce one bent constituent but it is sufficient to bend and dry the pre-compressed wood. Tokodi [19] used this modified wood in her thesis to design clothes and clothing accessories. Pleated wood can be used for vibration-dampening tool shafts and custom-shaped tools, picture frames, modeling, and saddles and other horse equipment. It can be used for furniture, interior design applications, construction, vehicle manufacturing, toys and sports equipment, musical instrument manufacturing, fine arts, and medical aids [9,16,18,20–22].

In the past decades, this material has been introduced in many workshops and conferences for craftsmen, artists, college students, and researchers across Europe and America. The technology is also described in modern literature (Dósa et al. [23], Brownell [24], Reinhard et al. [25], Lohmann [26], Brownell [27], and Smardzewski [28], among others). It is an industrially and scientifically known method. Many uses exist, as illustrated in Figure 1, without the need for completeness. Only the creativity of users can limit the diversity of bent products made from pleated wood.

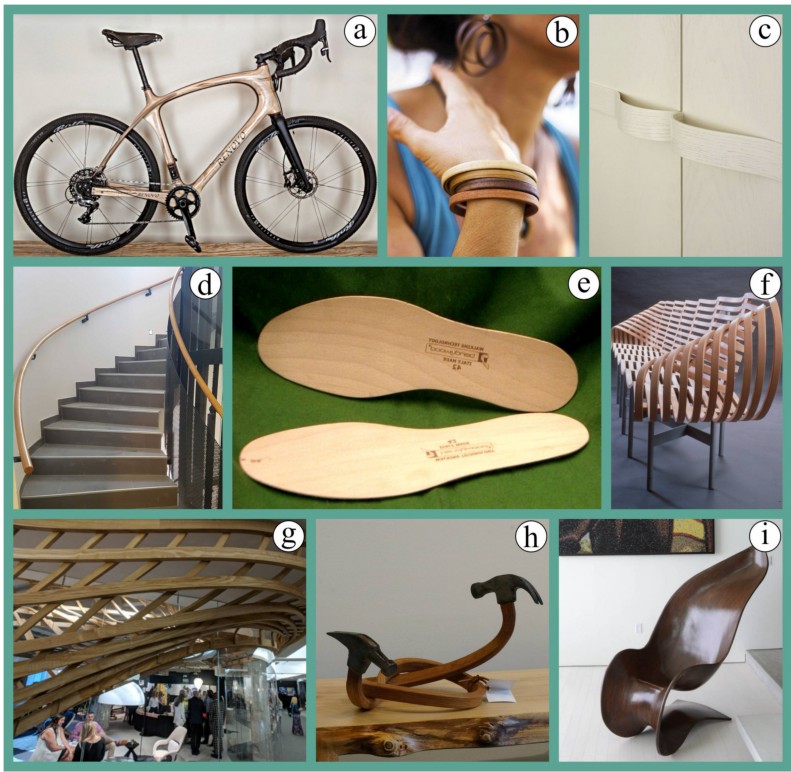

**Figure 1.** A few possible applications of pleated wood. The montage is made with the permission of the publishers of the reproduced or adapted images, cited in the References. All permissions were issued in 2023. (**a**) Pursuit bicycle (source: [29]); (**b**) Jewellery in Bendywood (source: [30]); (**c**) Bent wooden wardrobe handles (source: [31]); (**d**) Curved wood railings (source: [32]); (**e**) Wooden insoles for shoes (source: [33]); (**f**) Mobius bench (source: [34]); (**g**) 100 feet long organic trellis (source: [35]); (**h**) Hammer heads on bench (source: [36]); (**i**) Gulla chair (source: [37]).

Pleated wood, which can be bent in any direction, can be subjected to large deformation and can be processed with low waste [10,38]. No large manufacturing oversize is required, and the structure of wood remains intact [11] because the fibre direction always follows the arc. Its appearance is the same as that of untreated wood, but there is always some discoloration depending on the method and the duration of fibre softening. Due to its stockability and pliability, it is an outstanding raw material for series-produced bent constituents. It has a great advantage in the preparation of individual works, too [38]. Pleated wood is an excellent product from the point of view of aesthetics, material saving, and machining. However, even today there is still little awareness of the technology and its products. With expected future shortages of wood and labour worldwide, its role could become more important because it can be used quickly and with little loss of material for curved solid wood elements.

## 2. History of Pleating

The technology of wood bending using steam was already known by the ancient Egyptians, as evidenced by a tomb painting nearly two millennia before Christ [4]. Wood bending within the use of wood is about the same age as the culture of humanity. The demand for bent wood for different purposes has existed throughout history. Examples include the Greek klismos chair from the 5th century BC [3] and the British Windsor chair from the 18th century [39]. In 1808, Samuel Gragg in the USA patented a wooden chair containing bent parts [40]. There are also typically chemical fibre softening modes, such as cooking the wood in aluminous or ammonia solutions [16]. Impregnation with glycerin also improves the bendability of wood, but high temperatures are always required to achieve the desired result [16]. As an example, the advantage of ammonia treatment against classic

water steaming is that the treated wood remains plasticized even after the wood has cooled down until the toxic ammonia evaporates [41]. The German Michael Thonet was the first who bent wood by steaming in serial production in the Austro-Hungarian Monarchy from the mid-19th century. Initially, he experimented with thin-layered adhesive-based products, and then discovered the potential of heating wood in wet media. He reduced the structure of the chairs to a few components, making it possible to produce them in large series with cheap, unskilled workers [42]. In addition, the variation of the elements can be used to produce unique constructions on demand; in this way, Thonet reformed the furniture industry. His first patent was published in 1842, while the most important patent in terms of the manufacture of bent furniture was published on 7 October 1856 with the title "For the production of chairs and table legs made of bent wood, the bending of which is accomplished by the use of steam or boiling liquids" [43]. Steaming is only economical in large-scale production and it is difficult to carry out [16]. After the steamed wood has cooled down, it cannot be bent further [16,44]. For example, the raw material of the back legs of a Thonet No. 14 Vienna chair must be bent within a half-minute after the raw material has been taken out from the oven [45].

With the further evolution of industrial technology, it was possible to develop a process that provides flexible wood even at room temperature. This process is wood compression along the grain, first patented in the German Empire in 1917 [46]. The development brought new opportunities because it results in a highly flexible material that uses much less bending force than uncompressed wood, mainly in its wet condition but occasionally in its dry condition as well. Moreover, unlike steam-bent wood, compressed wood along the grain does not need heat during the actual shaping. By the end of the 20th century, it had become common practice to adjust the shape of curved elements on site [47]. According to Hanemann [46], large pieces of wood should first be cooked or steamed and then placed in a press in their hot and humid state, where they are surrounded by a cover during the process to prevent buckling. With the press plates, the wood is compressed in the longitudinal direction, then cooled and dried so the wood can be further processed (cut into boards or machined) and easily bent. With the same content, the Danish Pedersen was granted patents in several countries in 1918. The second patent of Max Hanemann [48] presents a more productive version of pleating.

Using a suitable device, the compressed length of the wood must be fixed and removed from the press in its compressed state and then cooled and dried. The latter operations can thus be carried out faster while the press can continue the production process. In the absence of adequate technological solutions, the method had not yet become known and spread. A few years later, in 1926, Holzveredelung Ltd. (Essen, Ruhr, Germany) [49] was the first to introduce industrial compression equipment, supplemented with technical drawings. The machine is designed to be able to compress a wood piece along its grain. The wood is pressed to the side wall during compression but it may move longitudinally along the wall (Figure 2). The clamping device lies under the press plate of the workpiece, can be fastened to the workpiece after compression, and keeps the wood in its compressed state. Taking the clamped wood out of the machine, drying will be easier and faster and production accelerates because the next workpiece can be compressed in the machine.

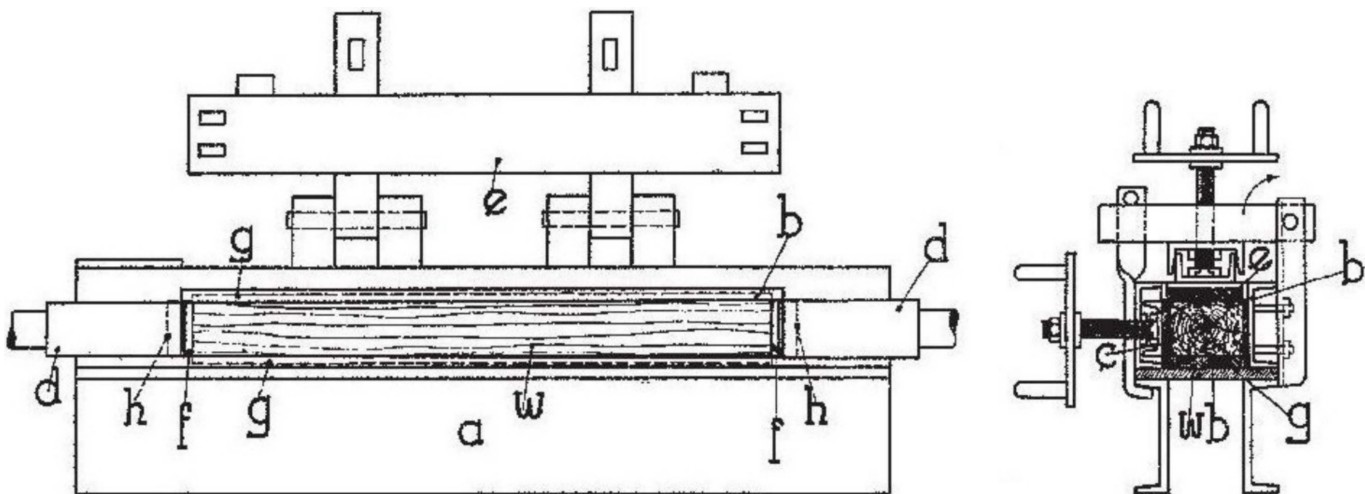

**Figure 2.** Longitudinal and cross-sections of the first compressing machine. Reproduced from [49]. Abbreviations: a—stand; b—groove; c—openable side wall; d—press plate; e—openable side wall; f—mobile end wall; g—mounting stem; h—groove; w—workpiece.

Magyar–Amerikai Plc in 1927 (Budapest, Hungary) [50] further developed the technology to be able to ensure an equal compression ratio along the entire length of long wood pieces by reducing friction between the wood and the side walls of the machine. During compression, high friction occurs on the side walls of the press, which hyperbolically reduces the compression force towards the center of the wood [16], i.e., consumes a large portion of the applied compression force. This is avoided by the support plates covering the inside of the press mould because the plates can move in the mould following the wood (Figure 3). It is also possible during compression to move tooth-surfaced support plates in the direction of the compression force. In this way, these plates transfer compression force to sections further away from the ends of the wood. The disadvantage of this solution is that due to its design, the tooth-surfaced support plate damages the surface of the workpiece. However, during the next manufacturing steps after pleating, the damaged surface is removed. Additionally, using the angle irons and fasteners on the corners, the compressed length of the wood can be fixed.

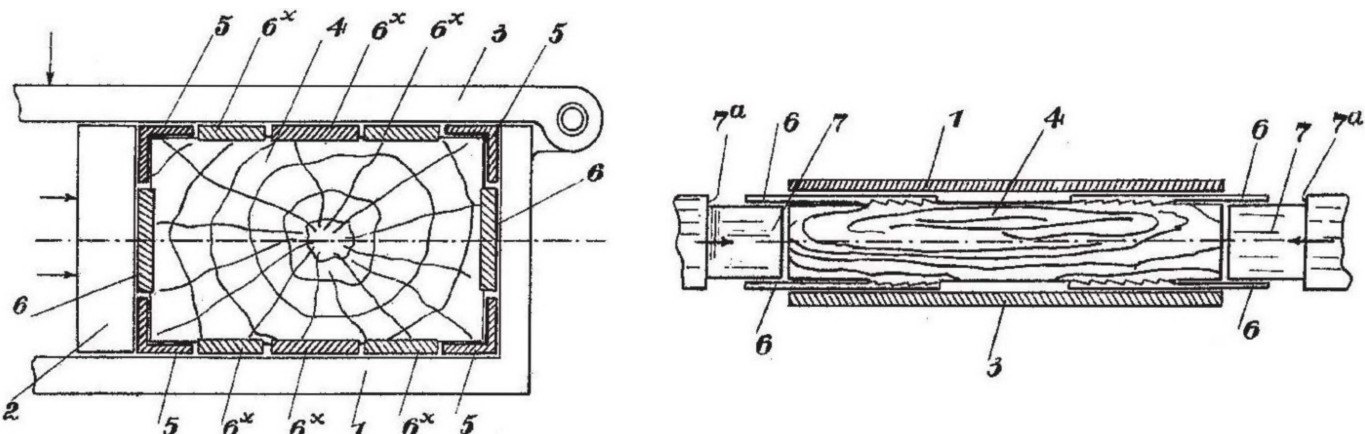

**Figure 3.** Sections of the compressing machine. Reproduced from [50]. The arrows indicate the direction of the forces. Abbreviations: 1—side wall of the press; 2—side wall of the press; 3—side wall of the press; 4—workpiece; 5—angle iron; 6—mobile supporting plates; $6^x$—serrated mobile supporting plates; 7—press plate; $7^a$—piston rod.

In the second half of the 1920s, the British Anglo-European Company Ltd. (London, UK) received numerous patents across Europe that are essentially identical to the wording

and drawings of previous patents of Pedersen [51], Holzveredelung [49], and Magyar–Amerikai Plc [50], translated into the language of the country where the patent was made.

According to August Thurn [52], the production of defect-free pleated wood with a rectangular cross-section results in a great loss of material because good-quality parts falling from the logs become waste. Using wood with a cylindrical cross-section, the quality of pleating is less affected by the defects in the fibre structure. The raw material must be placed in a support case, which prevents it from buckling, in which logs of various diameters can be compressed with an appropriate space-filling casing (Figure 4). Fibre softening and subsequent drying can be accelerated by removing the pith. During compression, this should be replaced by a pressure-resistant metal core. In order to speed up the drying process, hot compressed air can be circulated in the hole that remains after the pith is removed. According to the description, the production costs of the pleated wood are reduced to one-quarter and it becomes possible to produce peeled veneer of pleated wood. The technological details have not been patented and the method has never been used.

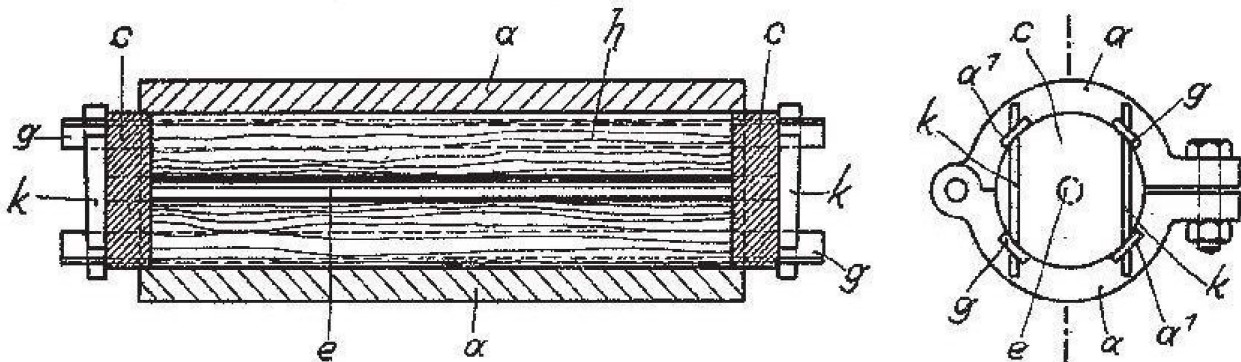

**Figure 4.** Sections of a machine that is able to compress cylindrical wood in its longitudinal direction. Reproduced from [52]. Abbreviations: a—case; $a^1$—groove; c—press plate; e—core hole; g—mounting stem; h—workpiece; k—fastener.

Thurn and Thurn [53] said that the greatest available flexibility is not always necessary, but using a lower compression ratio has not been a suitable solution. The authors have probably come to this conclusion because wood that has been less compressed results in both less equal compression along the length and in a less uniform pliability. It was known that after pleating, the wood can be cooled and dried in its compressed state, resulting in a very flexible material, whereas if the wood is allowed to spring back after pleating, it will be less pliable. By regulating the spring back, it is therefore possible to produce wood with a desired uniform pliability along the length according to its intended use. This will speed up the manufacturing process and reduce the costs of production.

Many written memories—such as the book by Graf published in 1932—provide evidence of the industrial production of pleated wood [54]. The book by Kollmann [55] on wood technology already introduces the characterization of pleated wood. Schneider [56] describes that pleated wood is used for aircraft components. Heisel and Eggert [57], Bátori [20], and Material Archive (Zürich, Switzerland) [58] link the first industrial production to Holzveredelung Ltd., so the industrial production of pleated pliable wood might have begun in the second half of the 1920s. For half a century after the 1940s, there does not appear to be any new patent or research in conjunction with pleating, only references to previous knowledge (Vorreiter [18], Blankenstein [59], König [60], Schietzel [61], and Heisel and Eggert [57], among others). According to the British Stevens and Turner [62], patents are already known for the production of pleated wood, but the technology is limited from a mechanical and economic point of view.

Chronologically, the next patent is owned by Sparke in 1989 [11] in Denmark, which deals with double compression. The author describes combinations of different process components (fibre softening, compression, quick or gradual release of the wood after

compression while it dries, various repetitions of these processes, fractional compression, etc.) and the possibility of variable side pressure in describing the technology. In addition, the patent specifies a generally used 20% compression ratio relative to the original length, and that in some cases, the process makes the wood permanently close to the plastic state, which remains even in dry conditions.

Thomassen et al. in 1990 [63] assembled a wood compression device. The text of the patent is supplemented with technical drawings of the patent granted in Denmark in 1989, which uses both pre-patented inventions and new technological innovations. The side wall components are pushed against the wood by a hydraulic tube and thus the side pressure can be different along the length of the workpiece (Figure 5). To the authors' knowledge, nowadays, electronically controlled compression equipment using the principles of this patent (manufactured by Compwood Machines Ltd., Slagelse, Denmark) are operating in Hungary and in the USA [5]. In addition, machines operating based on the patent of Magyar–Amerikai Plc [50] are used in Italy [20].

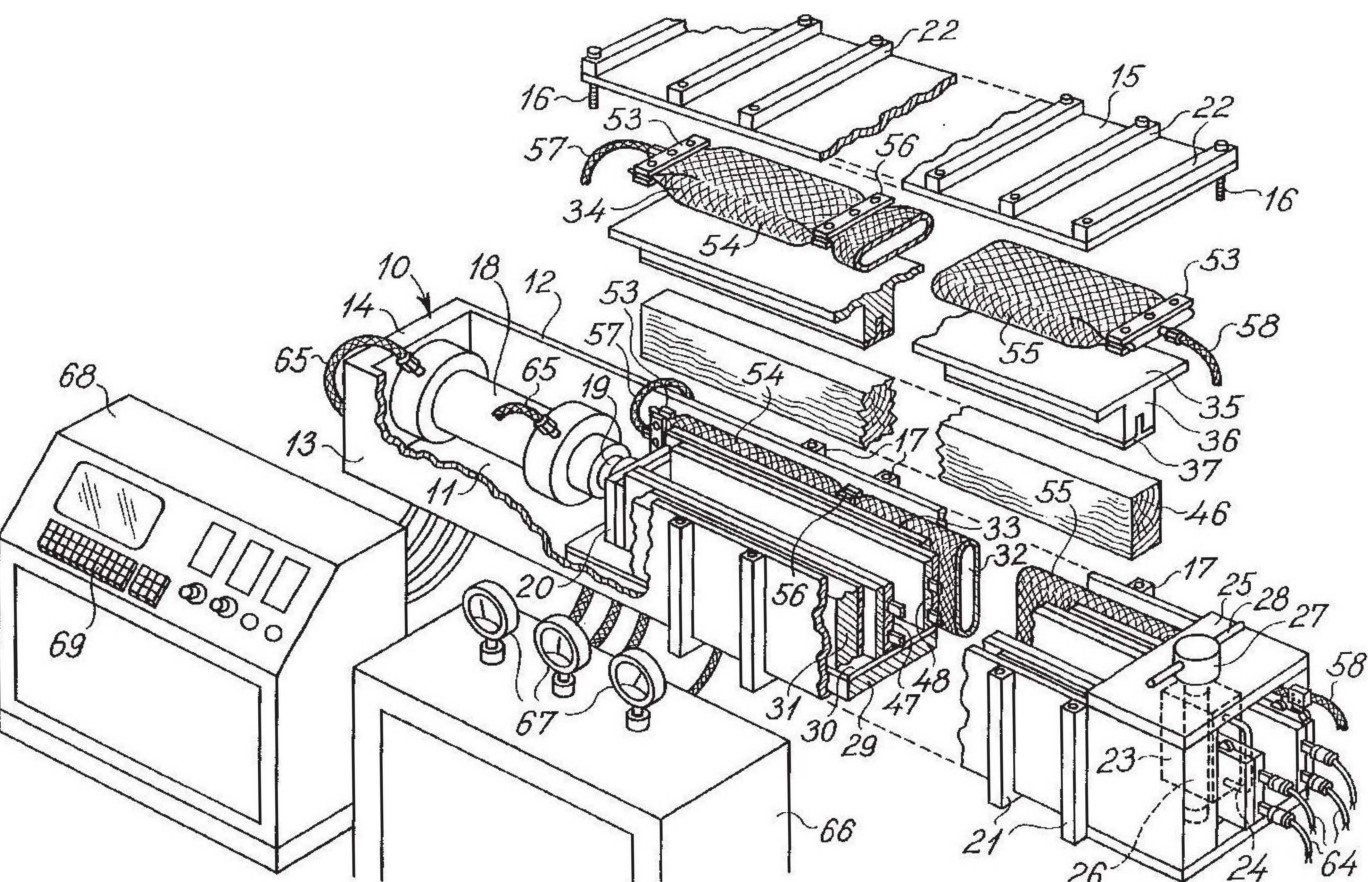

**Figure 5.** Theoretical scheme of a compressing machine. Reproduced from [63].

Bakos [64] describes an experimental device developed in Debrecen, Hungary. It used ball screw technology to compress rods made on turning lathe with a diameter of 22 mm. According to the description, this machine operated on the basic principles of the patent of Magyar-Amerikai Plc [50], but no further data on the equipment can be found, so the development was probably stopped. Chronologically, the following patent presents the usual process of preparation and pleating of wood, illustrated with a wood piece of 100 × 120 mm cross-section [12]. The advantages and possible uses of pleated wood are discussed. The machines for bending metals in wood bending also appear. Pleated wood can be used glued or mechanically fastened, and it is possible to make slight corrections of the bending radius during installation of the finished product. This patent does not introduce new technology or a new method, but it summarizes a lot of information.

Volkmer et al. [13] approached pleating from a new direction: a short free section of wood between two clamps is pre-compressed by about 5%. The free section always advances and another section is compressed until the entire length is pre-compressed. In the second cycle, using the technology described in earlier patents, the pre-compressed wood should be compressed up to half of its original length. If it is necessary to set the wood to an exact final length, it must be dried under pressure after compression. If the wood is compressed 10% shorter compared to its desired final length, the fastening devices required for drying under pressure can be omitted. A 5% longitudinal stretching of the longitudinally compressed and dried wood further reduces the stiffness, making bending even easier. Currently, no modified wood is produced under this patent.

A compressing device has been operating at the University of Sopron since 2015. The required pressure and continuous measurement data are provided by an Instron 4208 (Instron Corporation, Norwood, MA, USA) universal material testing machine. When the device is attached, specimens of $20 \times 20 \times 200$ or $20 \times 30 \times 200$ mm$^3$ can be compressed in the fibre direction up to 33% compared to their original length. The side walls of the semi-closed chamber are heated and able to move along with the specimen during the compression process if this is required [21,65].

It is worth mentioning additional patents loosely related to pleating. Curtis [66] cooked maple, beech, and birch raw materials with a moisture content of less than 12% for 2 h in crude oil, and then compressed them through a truncated mold after 2 days of drying. The thickness of the wood preheated to 100 °C was compressed to 84% in the heated press, and its length was also reduced while its width remained unchanged. He found that with longer cooking time, the wood becomes saturated and the required compression force increases. The purpose of this operation is to produce mechanical properties similar to dogwood (*Cornus*) in other wood species to produce a particular loom component, which is achieved by the combination of longitudinal and transverse compression after the softening of fibres. Jouko [67] discloses snap-on wooden constituents with reference to the original patent published in 1996 and identifies the pleated wood as a raw material, among others.

Szabó et al. [45] describe the main parameters of compression technology (moisture content, compression ratio, etc.) and a wooden spring made of pleated wood that can store and deliver energy. It can be used as part of mechanical, medical, furniture, or toy constructions. Eckardt [68] designed a means for demarcation of space using pleated wood for furniture parts, interior design elements, and decorative purposes. In a wood panel made of pleated wood there are several lines cut parallel and offset to each other and parallel to the fibre direction. Thanks to this, the wood can be laterally drawn out and made in a wide range of widths.

As is apparent from the patents presented, there have been numerous demands and technological problems over the past hundred years in the development of wood compression along the grain. Several solutions have been developed to overcome these problems, and nowadays, the production of high-quality pleated wood has been solved. It is possible to produce a variety of special products, primarily serving the needs of furniture and interior design, and further opportunities continue to arise. The aim of this review article is to introduce the raw materials, manufacturing methods, and cellular and mechanical changes of wood due to pleating based on scientific results.

## 3. Raw Materials

### 3.1. Wood Species

Buchter et al. [45], Kovács et al. [69], Compwood Machines Ltd. [70], and Rónyai [71] found the following wood species suitable for pleating: beech (*Fagus sylvatica* v. *Fagus* ssp.), silver maple (*Acer saccharinum*), black cherry (*Prunus serotina*), black walnut (*Juglans nigra*), ash (*Fraxinus excelsior*, *Fraxinus americana*), wych elm (*Ulmus glabra*), field elm (*Ulmus minor*), and oak (*Quercus* ssp., *Quercus petraea*, *Quercus velutina*). Kollmann [16] generally referred to deciduous wood species as suitable and specifically mentioned pear wood. Szabó [72] additionally listed robinia and birch, while Kuzsella et al. [73] listed robinia and linden

as compressible species. However, Buchter et al. [45] found that the following species cannot be pleated: birch, teak, mahogany, and rubberwood. According to Kuzsella [74], poplar is also not suitable. The machine manufacturer Compwood Machines Ltd. [75] listed cypress wood (North America and Europe), afzelia (North Africa and Southeast Asia), and pau marfim (South America) also as suitable for pleating, while Hyams [39] compressed Australian species mountain ash (*Eucalyptus regnans*), jarrah (*Eucalyptus marginata*), and spotted gum (*Corymbia maculata*). Pure Timber Llc (Gig Harbor, WA, USA) [76] works with North American wood species in addition to wood species found in Europe including red gum (*Liquidambar styraciflua*), red oak (*Quercus rubra*), white oak (*Quercus alba*), and sassafras (*Sassafras albidum*). König [60] and Sparke [11] only mentioned the hardwoods from the previous list, highlighting beech and oak, while Deibl et al. [12] and the Material Archive [58] mentioned a part of the European selection. Hyams [39] said that spotted gum can be compressed up to 12–15% depending on its origin. Experience shows that coniferous species cannot be pleated [18,74]. Kollmann [16] explained this by the significant difference in density and hardness of its earlywood and latewood. According to Ivánovics [77], the failure is caused by the thin-walled tracheid cell structure, which forms the earlywood of conifers and tends towards micro-bending and tearing.

### 3.2. Quality and Dimensions

The purpose of pleating is to make the natural wood pliable. Both pleating and bending require high-quality raw material. Straight-growing woods with narrow annular rings are the most suitable [65,69,72] (Figure 6a). Wood can be bent most ideally towards its heartwood side [45], and the direction of the grain must be parallel to the edges of the workpiece (a maximum deviation of β = 7° can be allowed) because the stresses during compression or bending may cause wood failures as a result of grain misalignment [72] (Figure 6b).

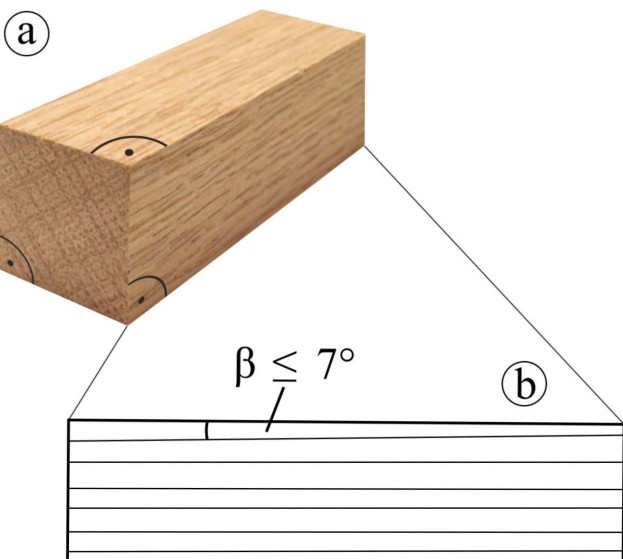

**Figure 6.** (**a**) The ideal wood structure for bending; (**b**) The allowable grain misalignment. The symbols with dots symbolise the right angles. Adapted from [59], with permission from Báder, 2023.

The type of sawing of logs does not matter from the point of view of the compression process. It can be plain sawn, quarter sawn, or transitively sawn between these two [45,70]. The ratio of sapwood and heartwood as well as the alignment of annual rings does not affect the quality of pleating [65,72,74,78] (Figure 6a). For beech and ash, false heartwood is not allowed [70]. In most cases, a fibre separation phenomenon can be observed when pleating wood containing false heartwood. The part with false heartwood always becomes damaged sooner, and occasionally, due to the local weakening of the material, the fissures spread to the clear area. Thus, the compressibility of wood along the grain is significantly reduced

by the presence of false heartwood [79,80]. Compared to the cross-section, relatively small knots do not affect the compressibility, but they can cause problems in bending, so it is recommended to use raw material without knots [45,65]. Fissures and insect holes are not permissible because they cause local weakening [65,70,79]. For serial production, it is advisable to pay attention to the origin of the raw material in order to create uniformly bendable parts.

The raw material may have any suitable cross-section using the appropriate support [63]. The shape and dimensions of the wood to be pleated are always determined by the compression device. Today's industrial equipment uses wood of various cross-sections (e.g., 80 × 120 mm, 100 × 120 mm) up to 3 m length [20,45]. It is also possible to compress several smaller wood pieces together into bundles, but only wood of the same species may be used and the bundle has to fill the compression chamber as if it were one piece of material. In a bundle, combining is possible both in cross-sectional and longitudinal directions. The surface of the raw material may be made by circular saw or by plane. It is important that the adjacent sides of the wood are perpendicular to each other (Figure 6a) [45].

*3.3. Moisture Content*

The initial moisture content (MC) of the raw material is an important factor for the compression process and can be divided into two main parts. Free water is found in the cell lumens, while bound water is connected to the cell walls. If wood contains as much bound water as it can, but the cell lumens contain no free water, this is called the fibre saturation point (FSP) [81]. The MC of a specimen is given as a percentage relative to its dry weight [82]. FSP of different wood species covers a wide scale; for example, robinia 19.5%, oak 24.5%, and beech 35.6% [83]. According to Buchter et al. [45], ideally, green wood is required for pleating, but it has to have an MC of at least 16%, which increases as a result of preceding plasticization to about 25%. It is very important to take into account that during steaming the MC increases the desired amount (bound water) and the wood swells and twists, so it might not be suitable for the modification process [45,79]. Furthermore, as a result of low MC, fibre separation may happen during pleating [79]. Kuzsella [74] found that wood with a moisture content of 2%–8% (MC%) less than its FSP is suitable for pleating. Szabó [72], Szabó et al. [14], and Rasmussen Ltd. (Helsinge, Denmark) [84] came to a similar conclusion and suggested that wood has to contain 20%–25% MC. According to Sandberg and Navi [3], an MC of 25% is sufficient. Deibl et al. [12] propose green timber for pleating, whose MC decreases near to FSP during steaming. The MC of the raw material should be at least about the FSP [65,79]. Kuzsella [74] tried to compress robinia with 32% MC unsuccessfully, but with an MC of 20%–25%, both 7.5% and 15% compression ratio were feasible.

For pleating, high-quality and clear deciduous wood is required with a uniform structure, which will be bendable after the treatment in the radial and longitudinal planes as well as in the tangential and longitudinal planes. Fissures, curved growth, false heartwood, knots, spiral grain, uneven edges, wormholes, etc. are failures that prevent compression. The literature suggests the moisture content required for compression should be around the FSP. It should be noted here that hardwoods have quite different FSPs.

## 4. Production of Pleated Wood

*4.1. Plasticization*

The first step in the pleating process is plasticization. For easy understanding of the whole process, Figure 7 shows a schematic overview of pleating, including posttreatments.

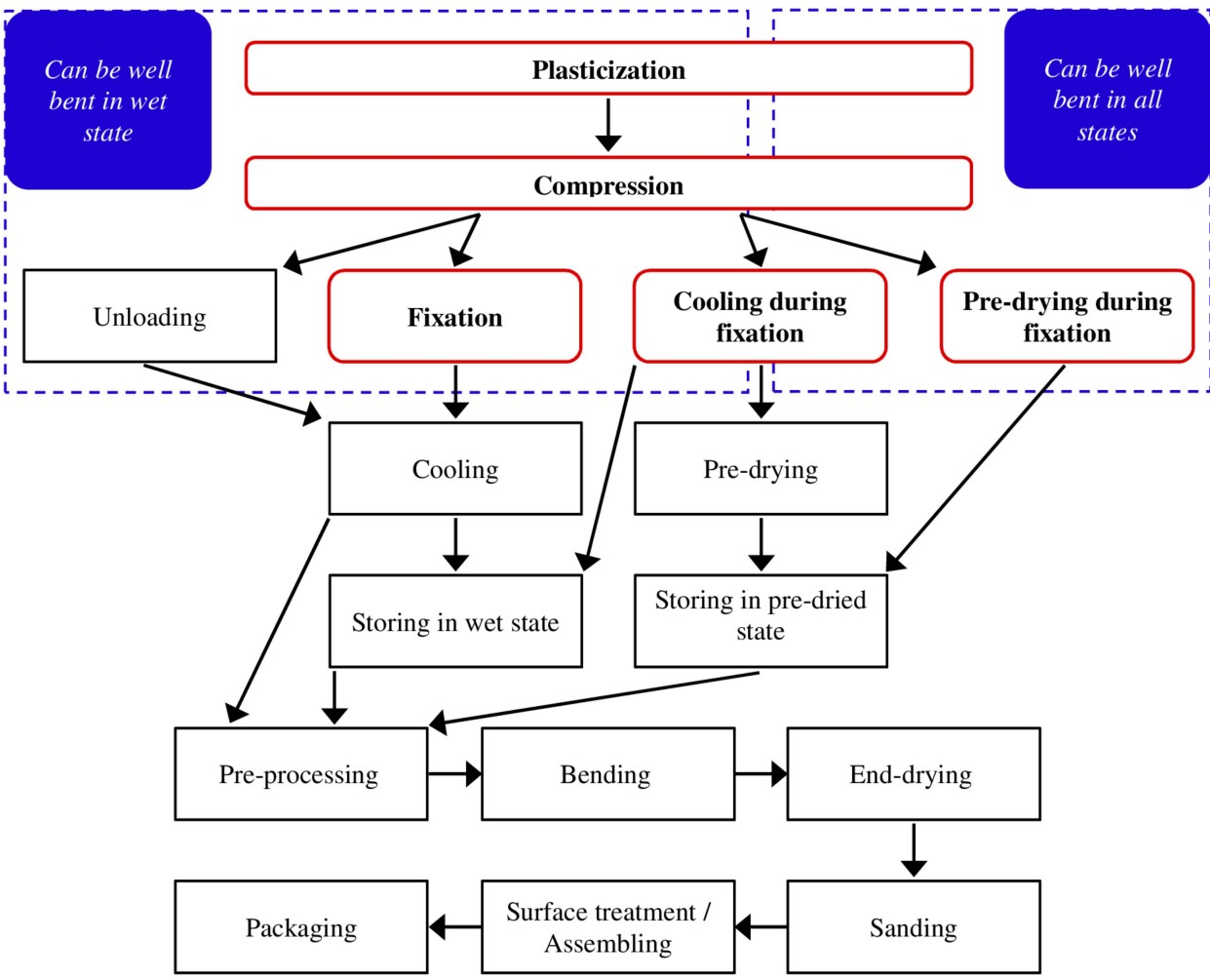

**Figure 7.** Posttreatment processes after pleating in use today. Reproduced from [59], with permission from Báder, 2023. The three main steps (plasticization, compression, and fixations) are highlighted using red boxes and bold characters.

Compressibility of wood along the grain is ensured by the hollow structure of wood cells. The higher-density deciduous wood species can be compressed in their plasticized state, and in this way, their structure changes but does not deteriorate [85]. The treatment comprises three stages: plasticization, pleating, and posttreatment. Wood is a hygroscopic material because it can both release water and take it up depending on the surrounding circumstances [86]. Both hemicellulose and lignin are present in wood tissue in an amorphous state, and the cellulose fibres are surrounded by these constituents [87], just as the concrete surrounds the iron reinforcement in a cement stab. When wood has a high MC at room temperature, semi-crystalline cellulose and hemicelluloses are in a rubbery state, but lignin remains in its glassy state; therefore, wood cannot be deformed easily [3]. Hemicellulose and lignin are more susceptible to thermal- and moisture-induced softening than cellulose, so increasing the temperature and moisture increases mechanical anisotropy [87]. Wood has to be plasticized before compression to avoid failures during the process [10,88]. The structure of the hemicelluloses and lignin, which hold together the microfibres and the cell walls, loosens [89,90] and the cellulose microfibres that give the stiffness to wood are displaced relative to each other during bending [69,91] and during pleating. According to Németh [92], cellulose is most resistant to hydrothermal treatment. Heat and moisture cause the break of the hemicellulose–cellulose and lignin–cellulose bonds in wood [90,93] at temperatures below 100 °C [94,95]. Increasing MC decreases the softening temperature

of lignin and hemicelluloses [96]. At around 70%–75% MC, the hemicelluloses are already softened at room temperature [97,98].

During steaming, the decomposition of wood begins at temperatures below 100 °C, especially with the degradation of hemicelluloses, which reduces the resistance of the wood to pressure [90,99]. According to Yang et al. [100] and Poletto et al. [101], wood decomposition starts at about 70 °C in wet conditions. Vorreiter [102] says that after steaming at 90 °C, the residues of hemicelluloses are already present in the condensed water, and according to Winkler [93] and Sandberg and Navi [3], lignin becomes plastic at 85–90 °C in the presence of water. Sitkei [103] specifies 50 °C for hemicelluloses and 90 °C for lignin as minimum softening temperatures.

The maximum plasticity of wood can be reached at 25%–30% MC [69] at a minimum temperature of 80 °C for beech wood [96] and at 70–100 °C for pine wood [104]. After cooling or drying, the cell-linking middle lamella solidifies again [69]. In practice, generally, saturated steam at a temperature of 100 °C is used at atmospheric pressure to avoid desiccation during heating of the wood before pleating [3,44,84,85]. The time of plasticization must be sufficient to heat the entire cross-section of the material, which can be calculated as about 2 min per mm thickness [62,78]. Prolonged steaming and higher pressure have not shown better results [80]. Holzveredelung Ltd. [105] makes suggestions to accelerate plasticization before pleating, which can be achieved by a vacuum. Holzveredelung Ltd. [105] further states that by applying a vacuum, the different softening of the wood caused by different structural irregularities will be substantially reduced. Consequently, the pleated wood will be more uniformly bendable. Thomassen et al. [63] show that partial compression of wood can be achieved either by partial fibre softening or by partial variation of heating during compression. In summary, wood is generally subjected to steaming along its whole length, and a minimum temperature of 80 °C is required throughout the length and cross-section of the material during both plasticization and pleating.

### 4.2. Compression along the Grain

The goal of pleating is to avoid changing the cross-section and bending or breaking the wood [44]. During this process, the wood must be laterally supported to prevent buckling [59,72]. Gradually reducing the pressure after compression, the elastic deformation is equalized in the material, resulting in spring back, and a remaining shortening can be observed depending on the applied compression ratio. During an ideal compression, the deformation along the length is of the same magnitude, with smaller variability at higher compression ratios, and the failure of the wood does not begin in the weakest point. More compression stress is required for higher compression ratio. As such, the less compressed parts that require more compression stress to achieve a given compression ratio will also be compressed to the ratio that has already occurred in the other parts as a result of the continuously increasing stress [77]. The used compression ratio is usually 10%–30% compared to the original length of the specimen [14,17,22,45,72]. This ratio is 10%–25% according to Ivánovics [77] and 10%–20% according to Vorreiter [18]. Sparke [11] and Sandberg and Navi [3] clearly give a compression ratio of 20%. The maximum available compression ratio is up to 25%–28%, followed by unwanted damage of one or more locations, and finally the wood is destroyed [77]. This was proved by Szauer [106], who examined samples of green sessile oak from different growing sites. The maximum compression ratio of oak sapwood from Sopron, Hungary is 27%–28%. Higher compression ratios generally result in increased remaining shortening with higher deviation. The maximum compression ratio of mature wood of oak from Sopron, Hungary is 21%–23%, while that of wood of oak from Zala, Hungary is only 17%. Szauer [106] achieved a much higher compression ratio of 30% for beech, also with at least a 90% yield of specimens without failures during the process, which is an exceptionally good yield. The compression ratio of softened wood depends mostly on the wood species (cell structure, density, etc.) and the requirements for the final product [63].

The rate of compression along the grain is important for both its productivity and quality. A new unit of measurement is needed to be able to compare objectively the different raw material lengths and applied operating times, used both in laboratory and industry. Báder and Németh [80] suggested a new unit of measurement, the relative compression rate $\left[\frac{m}{m \cdot h}\right]$, which ensures the comparison of the different pleating techniques. It represents the shortening that occurs on the workpiece per unit length over a unit of time. In other words, the relative compression rate explains how much shortening per meter of the workpiece occurs in an hour using basic units of measurement. Thus, relative compression rate is able to handle uniformly the substantially different basic data and is independent of the compression ratio [80]. A further simplified version of the introduced unit of measurement is [%/min] proposed by Báder [107], which will be used in the following. Its meaning is the percentage of the specimen compressed per minute compared to its original length. According to industrial sources, compression rates of 0.7%–4.0%/min are used [17,20,45,84,108,109]. The general compression rate of the laboratory equipment used at the University of Sopron, Hungary is 15.0%–25.0%/min [80]. The large difference (roughly an order of magnitude) may be due to the different compression technologies and the different cross-sections of the raw materials.

The highest pressure required for 20% compression of wood can reach 35 MPa [63], while Dienes [108] mentions a pressure of 25 MPa for the same machine type. Bátori [20] recorded a maximum pressure of 42–50 MPa for the equipment operated in Italy, while a 12–20 MPa pressure was applied to the laboratory machine depending on the rate and the wood species by Báder and Németh [110].

### 4.3. Posttreatment, Drying, Machining

Posttreatment is the last stage of the modification process, which can be performed many ways (see Figure 7). Once the desired compression ratio is reached and the forces are immediately eliminated, the still-plasticized (hot and wet) wood increases in size. Finally, it has only a slight permanent shortening of 3%–5% (Bátori [20], Szabó et al. [14], and Báder and Németh [65] say 3%–10%) which is the degree of compression and depends on the properties of the wood and the percentage of the initial compression ratio [3,4,14]. This phenomenon is called "spring back." Báder and Németh [111] found that most of the spring back occurs immediately after releasing the pressure, followed by a further minor spring back in the first three minutes. For oak, the total remaining shortening was 3.84% [111]. The spring back of pleated wood is caused by the internal stresses. Based on laboratory observation described by Báder [107], bending and straightening a pleated specimen several times in its wet and hot state after compression increases its remaining length compared to normal spring back. This is the effect of the buckled cell walls being partially straightened by the bending and straightening of the specimen. If the wood is held in its compressed state for a while following the compression—which is called fixation—and allowed to cool down, the wood retains its flexibility much better and, thus, the remaining shortening will be much larger [112]. If wood is dried and cooled during fixation, the entire shortening remains (right side of Figure 7) and the treated wood will always be pliable [12,63,69,75]. According to Sparke [11], this finished raw material can be stored at 6%–8% MC, and according to Bátori [20], it can be stored at an MC of 15%. These values depend on the wood species. In this condition, biotic organisms no longer cause problems. The relatively time-consuming compression process can be perfectly integrated into the industrial production line considering the properties of the method and the product (the pleated wood).

After compression without fixation for a long time, the wood is immediately suitable for bending, but it can also be stored as needed [11]. While its MC is over 20%, it retains its pliability [14,22,70,77]. Others give 15% MC [16,18,38,57,60], while [5,20,45,72,102,113] give 25% MC. According to Graf's book, published in 1932, the manufacturer delivered pleated wood with 39% MC to its customers [54]. Additionally, Vorreiter [18] says that pleated wood with over 15% MC has nearly plastic properties. Decreasing MC rapidly reduces

its bendability and, between 0%–5% MC, the treated wood is more brittle than untreated wood, so it is no longer pliable. Investigations by Báder and Németh [113] have shown that the pliability of pleated wood is best when it is close to its FSP; as such, 20%–30% MC is recommended, depending on the species. Drying of pleated wood can be prevented for up to 12 months when stored at the freezing point or for up to 6 months when it is stored in a dark room in plastic foil [45,70,72]. Wood should be protected from mould and fungi that can cause discoloration and other damage [44,45]. Pleated wood stored in wet conditions does not tolerate direct sunlight [44,70]. Pleated wood solidifies in its desired shape after bending, fastening, and drying [44,63,72]. It is advisable to perform bending to very small radii immediately after the compression process while the wood is still hot [38]. After steaming and drying, wood keeps the shape of the bend, but by re-moisturizing the finished bent wood, it recovers much of its original shape, called "memory effect" [114]. The memory effect as well as the shrinkage and swelling of pleated wood will be discussed in Section 5.2.

There are significant differences between the treatments shown on the left and right sides of Figure 7, but there are transitional solutions as well that improve the results of compression with less time and energy investment. According to Bátori [20] and Sőregi [17], with the industrial technology used today, the 3 m long board is compressed to 2.4 m and then allowed to spring back to 2.8 m, thus improving the strength values of the semi-finished product. According to another currently used industrial method, compressed wood is dried, cooled, and then conditioned in its compressed state [20]. Thereafter, this type of treated wood will be suitable for use. Many other versions of pleating can be combined from the two methods described above, but there is no available information regarding whether such forms of differentiated industrial production exist. A good example of an interesting combination is the method described by Sandberg and Navi [3], which suggests that after the wood is compressed by 20%, released, and dried to 12% MC, the specimen is subjected to a further compression of 20% in its cold state. After treatment, there is a residual deformation of about 15%, resulting in a highly increased bendability of the wood.

After compression along the grain and posttreatment, the wood piece can be machined and bent to the desired shape in a cold state, end-dried, and surface-finished as a straight piece of material [3]. If treatment is successful, no visible damage (e.g., fissures, disintegration) can be observed on the specimen. Bending can be performed with simple tools and jigs that are easy to use [10,72]. The bending of the pleated wood with small cross-sections can be solved freehand, and when larger parts are produced, it is necessary to use tools and jigs from Thonet technology that have been known for a long time. When wet, the pleated wood can be easily machined by turning and sanding, but it is worth planing perpendicular to the fibre direction due to the risk of torn grains [45,113]. Furthermore, during machining, strict guidance is required because of the flexibility of wet pleated wood [17,22]. When pleated wood is cleft, the messy state of the fibres can be observed [18] (Figure 8).

During the utilization of pleated wood, which is bendable in its wet state, it must be fixed to the correct shape (by nailing, screwing, clampings, etc.). Then, its MC must be reduced below 16% by means of a drying technology [16,38]. Vorreiter [102] gives 5%–8% MC and König [60] gives 4%–5% MC after drying, which must be conditioned later according to the climate of the place of use. According to Compwood Machines Ltd. [70], drying is generally carried out at a temperature of about 60 °C. Heisel and Eggert [57] give 60–70 °C for a few hours, while others give 80 °C [4,54,59]. Kollmann [16] and König [60] describe drying at 70–80 °C, whereby wood retains its set shape until its moisture content is below 15%–18%. When drying at temperatures above 80 °C, wood can easily be warped [16]. Pleated wood can withstand higher drying temperatures than untreated wood without the appearance of fissures [22]. Pleated wood adapts to new climatic conditions faster at first desorption, so the drying process of treated wood can be performed in less time [86]. Deibl et al. [12] suggest that the drying time is reduced by 5% compared to untreated wood. The MC of pleated wood coming off the production line is

usually below its FSP, so the drying temperature can be high from the beginning, reaching a high temperature quickly but in steps to prevent surface fissures. The high 60 °C initial air temperature can be further increased up to 80 °C. The moisture adsorption of pleated wood is slightly slower compared to untreated wood. Therefore, wood in use shrinks and swells less in varied climate conditions, providing a better cross-sectional dimensional stability and making the wood more resistant to drying fissures [86]. After drying and cooling, any pleated wood can be machined as untreated wood. This is true for sawing, milling, sanding, and surface treatment [11,38,48]. Heisel and Eggert [57] and Buchter et al. [45] described the same for gluing and surface treatment.

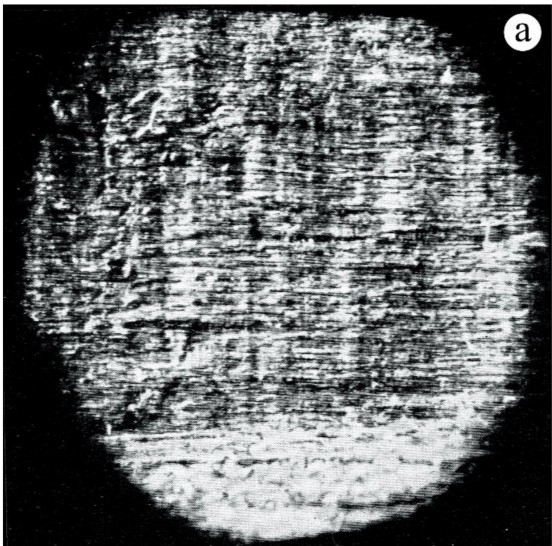 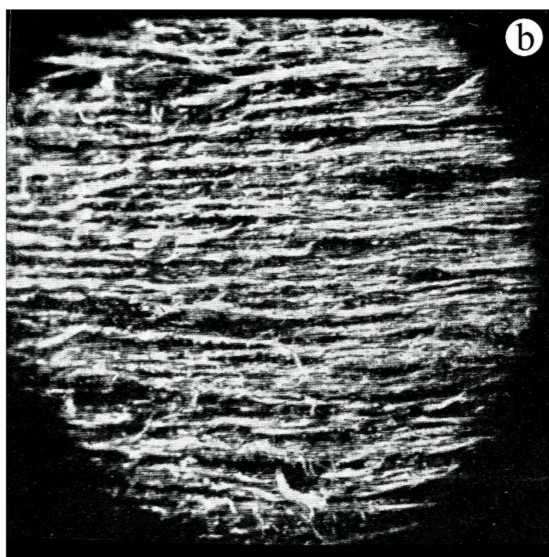

**Figure 8.** Images on the cleft surface of (**a**) untreated and (**b**) pleated beech wood. Reproduced from [59], with permission from Báder, 2023.

Due to plasticization and drying, wood is subjected to a certain degree of heat treatment as a result of the technology. As such, it complies with the requirements of the current ISPM 15 certification, ensuring the norms for international shipping. As the MC of the product increases, shape stability problems occur; therefore, this modified wood is not suitable for outdoor use [115]. No environmentally harmful substances are required for the production or use of pleated wood [12,72], so the product is free of chemical additives [116]. However, if the required product cannot be made of one piece of pleated material due to its size or geometry, it can be joined using, for example, tenoning. In this case, glue is added to the final product, but in a much smaller amount compared to conventional multilayered or laminated curved products [115]. The book by Conradsson [113] provides useful information for designers and manufacturers, summarizing the uses of pleated wood and the tools and devices for machining.

## 5. Properties of Pleated Wood

The mechanical properties of wood are influenced by many circumstances. In order to be able to evaluate and compare the results, these circumstances must be known. Mechanical properties of pleated wood can be found in advertisements, product information, and technology descriptions, but these typically do not contain sufficient additional information about the circumstances of the tests; as such, they do not necessarily reflect the position of science. Therefore, data originated from industry are not taken into account in this section. Some publications examine the effect of different ratios of compression on mechanical properties. For the sake of consistency and transparency, the properties of wood compressed by 20% are described. MC greatly influences the properties of the wood; therefore, MC is indicated where possible. In addition, wood species, wood quality, dimensions, and the

tools that are used determine what shape and to what extent the pleated wood can be bent, i.e., its properties, are indicated [10].

*5.1. Mechanical Properties in Dried State (Modulus of Elasticity, Modulus of Rupture, Tensile Strength, Compressive Strength, and Impact Bending Strength)*

The most important property of pleated wood is its modulus of elasticity (MoE), which has a strong correlation with the bendability of the material [117]. Thomassen et al. [63] and Szabó et al. [14] say that pleating reduces the MoE of wood. The components of the tensor of stiffness are slightly reduced [14]. Other authors provide quantified data, as shown in Table 1.

**Table 1.** Changes in the modulus of elasticity (MoE) and the modulus of rupture (MoR) of beech and oak due to pleating. Moisture content (MC) during the test is important; thus, it is listed in the table where possible.

| Beech | MoE | MoR | MC |
|---|---|---|---|
| Vorreiter [18] | −75% | −45% | - |
| Kollmann [54] | −86% | - | 18% |
| Blankenstein [59] | −86% | −55% | - |
| Buchter et al. [45] | −15% | −10% | 5% |
| Ivánovics [88] | −68% | −46% | - |
| Kuzsella and Szabó [44] | −70% | −47% | - |
| Báder and Németh [110] | −66% | −47% | - |
| Báder and Németh [112] | −46% | −19% | 12% |
| Somogyi [118] | −55% | −24% | 9% |
| **Average** | **−63%** | **−37%** | **11%** |
| **Oak** | | | |
| Kollmann [54] | −86% | - | 18% |
| Blankenstein [59] | −85% | −55% | - |
| Ivánovics [88] | −60% | −42% | - |
| Kuzsella and Szabó [44] | −45% | −17% | - |
| Báder and Németh [110] | −59% | −3% | 12% |
| Báder et al. [119] | −63% | −5% | 12% |
| **Average** | **−66%** | **−24%** | **14%** |

As a result of 20% compression, the MoE reduces to one-third of its original value, on average, for both beech and oak. For maple, this change is smaller [59]. The pleating of beech wood does not affect its MoE determined by the spreading velocity of ultrasound, either in the radial or in tangential directions [74]. However, both static and dynamic elastic moduli in the fibre direction are reduced to one-third of their original value, while the modulus of elasticity determined by the natural frequency of vibration decreases by a third of the original value [74]. The modulus of rupture (MoR) of beech at 20% compression ratio is reduced by more than a third compared to untreated beech, while the change is smaller in oak. During the bending test, the maximum deformation until fracture both for beech and oak wood increases approximately three to four times compared to untreated wood, but the specimens do not always fracture [44,112,118]. Depending on its duration, fixation can double the bendability of wood [9]. The areas under the stress–strain curves increase fourfold, which shows the strengthening of the toughness [44,73]. According to Szabó et al. [14], the work needed to fracture beech wood increased threefold due to pleating. With pleated wood, a reserve of strength remains even for high deflection [14,54]. This is evidenced by the graphs in Figure 9, which show the bending results of beech wood compressed by different compression ratios. Segesdy [15] investigated the differences between the mechanical results of the radial and tangential directions of pleated beech wood. The differences between the anatomical directions in dry state were 7.5% for MoE and 5.8% for MoR, in favor of bending in the LR plane.

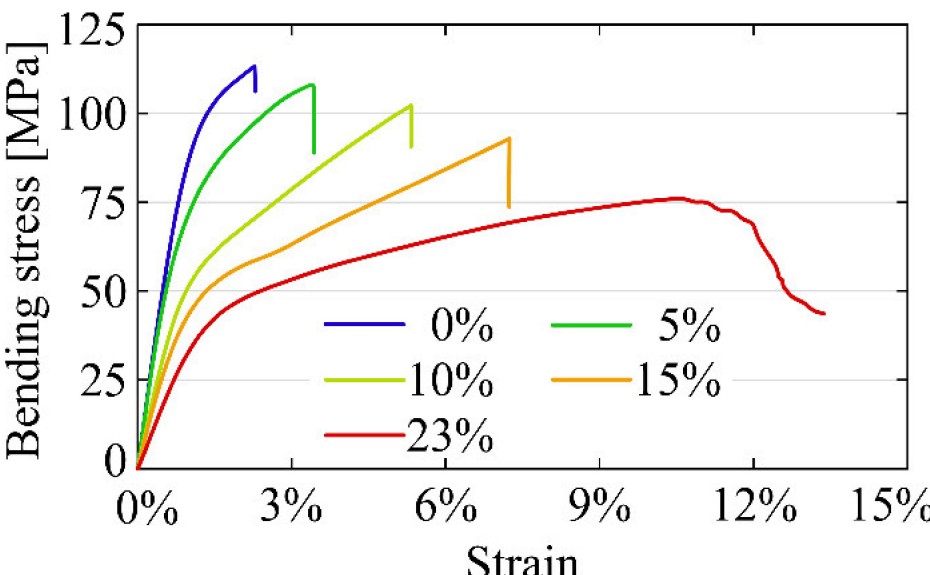

**Figure 9.** Stress–strain graphs of the 3-point bending test data, which represent the investigated compression ratios well. Numerical test results of Kuzsella [74] were used to plot the graph, reproduced from [59] with permission from Báder, 2023.

As a result of 15% compression of beech wood, its tensile strength is reduced by 36%, and its tensile modulus is halved, while the elongation at fracture increased by 96% [120]. This also shows that, as described by Báder et al. [121], pleated wood can withstand a much greater deformation on the tension side during bending. According to Heisel and Eggert [57], the elongation of the convex face during bending of wood is possible to the extent of the degree of compression. We will return later to the explanation of this phenomenon as well as the explanation of the 5%–10% higher force needed to split the pleated beech wood; see Section 5.4 for both [18].

The compressive strength parallel to the grain is reduced by an average of 7% compared to untreated wood [18,45,112,122]; however, at the fracture points, strong buckles always appear perpendicular to the direction of pressure [54]. During compression along the grain, the wood has been compressed beyond its elasticity limit in plastified state. The proportional limit obtained during the compression test for pleated beech wood is about two-thirds of the untreated sample, and in many cases the elastic range cannot be determined [112]. Báder [107] found similar trends in oak sample. The compression without fixation approximately halved the compressive modulus of elasticity to 2 GPa. This value remained stable using short fixation periods (1 and 3 min). Fixation for a long time (18 h) minimally increased the compressive strength and reduced the compressive modulus of elasticity to about one-fifth of the original value. The compression tests parallel to the grain of Grászli [122] resulted in an apparent reduction in the length of the pleated and pleated heat-treated beech and oak samples, with much less spring back compared to the untreated sample. In terms of lateral Brinell hardness, Grászli [122] found a 10% increase for beech and a 25% increase for oak as a result of pleating, while the decrease in end-grain Brinell hardness was 5% for both species.

Kuzsella [74] found that as a result of compressing beech by 23% along the grain, the specific impact work (Karman critical velocity; KCV) doubles. According to Szabó et al. [14], KCV increased by 68% for beech. According to Ivánovics [88], KCV increased by 191% for beech and by 73% for oak. Similarly, after 20% compression, Kuzsella and Szabó [123] found a great increase in KCV: 84% for beech and 86% for oak. The sustained deformation, the impact bending strength, and the KCV (the toughness of the wood) are significantly improved by pleating [74,123]. The value of the acoustically investigated logarithmic decrement increases up to four times, which proves that, as a result of pleating, beech wood exhibits a more viscoelastic behavior [74]. Two far-reaching statements from

Kuzsella [74] that both the determination of the spreading velocity of ultrasound and the determination the MoE from the natural frequency of the logarithmic wave may be suitable for determining the degree of compression. These methods can also be of great help in checking uniform compression along the length of a piece of wood. In addition, pleating generally increases the coefficient of variation of the measurement results [9,74,88].

*5.2. Physical and Related Properties in Dried State (Bending Ratio, Density, Memory Effect, Shrinkage–Swelling, Sorption, and Chemical Changes)*

The maximum possible deformation is most easily determined by the smallest bending radius measured on the inner arc, which depends on many factors (mainly the cross-section of the specimen, wood species, material quality, degree of compression, jig, and tool used for bending). All publications on the subject agree that pleating significantly reduces the minimum bending radius and the bending force. However, even for pleated wood, large forces are required for bending specimens with large cross-sections [72]. Both compressibility along the grain and bendability depend on the cell structure and therefore vary by wood species [72].

The bending ratio, which is also referred to as the bendability coefficient (h/R ratio, or $k_{bend}$), is a size-independent parameter of the material. Based on $k_{bend}$, the tightest bending radius available (R) can be calculated for a given thickness (h), which is technically safe to use. R is the radius at which fracture occurs in not more than 5% of the wood samples [124]. $k_{bend}$ specifies pliability and energy absorbing capability at the same time and it is inversely related to elasticity. By increasing the compression ratio, the $k_{bend}$ can be improved [74]. Figure 10 shows the bending ratios available for beech at room temperature in a wet state.

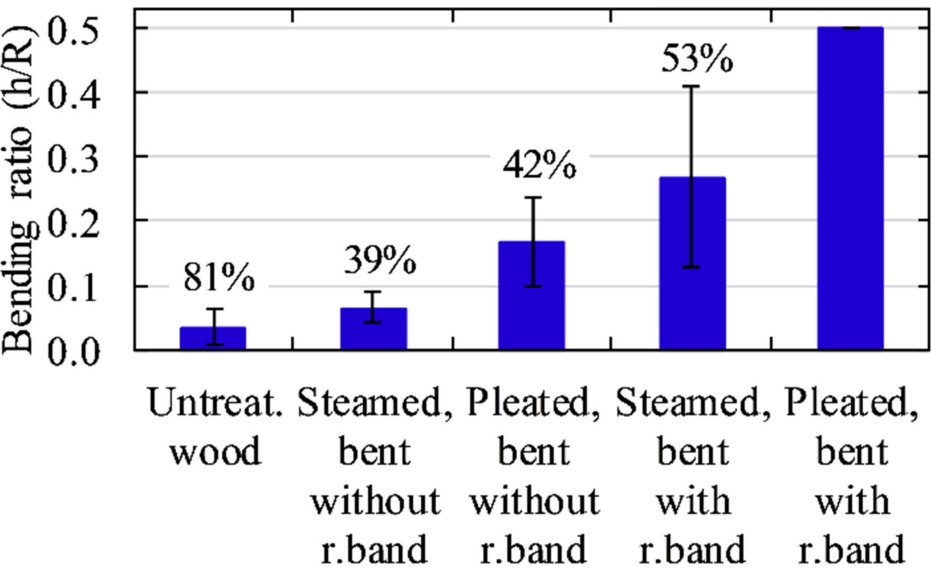

**Figure 10.** Average bending ratios and standard deviations of 25 mm thick beech wood using different bending processes; coefficient of variations are shown above the diagram columns (The numerical data used to plot the diagram are taken from the following sources: [3,4,12,14,16,38,57, 64,74,102,108,124,125]. Reproduced from [59], with permission from Báder, 2023. Abbreviations: untreat.—untreated; pleated—compressed along the grain; r.band—restraining band.

The differences between the raw data of the averages of $k_{bend}$ shown in Figure 10 can be attributed to experiments carried out with different settings and tools on differently structured beech samples. However, they are uniform in that pleating significantly improves bendability. After compression, better bending ratios can be achieved compared to the classical steaming process, and the pleated wood can no longer be formed only in its plasticized (hot) state. Curves that earlier could only be produced in a hot condition using a restraining band can be made using pleated wood at room temperature, even without a restraining band. The substantially higher $k_{bend}$ expands the usage possibilities.

It is also worth mentioning that Dienes [108] found significant differences between the bendability of materials with rectangular and circular cross-sections. According to a study by Vorreiter [102], with increasing thickness, $k_{bend}$ slightly decreases. Ring-porous wood species can be more easily shaped than diffuse-porous species.

The double compression of wood in the longitudinal direction [3] is not used today. As an example, after this treatment, 1 coll (25.4 mm) thick beech wood can be bent in its cold state to a bending radius of 100 mm without the use of a restraining band, and thus a much better result can be achieved with this treatment.

During the four-point bending test after conditioning (20 °C and 65% relative humidity), some of the fibres on the tension side break, but the intact parts hold the specimen together, allowing it to spring back after the load is removed. Based on the work of Grászli [122], the immediate spring back of an untreated sample was 58% compared to the maximum deflection measured during the bending test, while the spring back measured 5 days later was 65% for beech and 60% for oak. For the pleated sample, the results obtained immediately after the bending tests were 25% for beech and 32% for oak, while the results obtained 5 days later were 37% and 35%, respectively. More stress remained in beech after bending, and the pleated samples, despite their 2.5 times higher maximum deflection, only sprang back half as much due to increased ductility.

According to Kuzsella [74], the changes in the length and in the cross-section after the compression approximately equalize each other independently of the ratio of compression; as such, the change in density is not significant. Despite this, Báder and Németh [9] stated that fixation for a long time after compression along the grain minimizes the spring back of the specimens, while the cross-section of the specimens increases only minimally. In this case, a higher-density wood is obtained after treatment. Based on the densities calculated from the beech and oak shrinkage–swelling tests by Horváth [126], no clear difference can be detected between samples that were untreated, pleated, and fixated for a long time. According to Szauer [106], pleating does not change the density, and after fixation it is increased only slightly, by up to 10%.

Pleating of wood by 20% produces three to six times greater shrinkage in the longitudinal direction during the drying process compared to the untreated sample, depending on the fixation time [9]. The shrinkage–swelling tests by Horváth et al. [127] were carried out on beech and oak. The results of the tests have shown that the length of pleated wood increases during the soaking and drying periods, so that the longitudinal swelling values at the beginning can be very high, up to 10%, which negatively affects the usability of pleated wood. This is an outcome of the memory effect, which has to be separated from the three other factors: spring back, remaining shortening, and shrinkage–swelling. Báder [107] conducted a more in-depth study of this phenomenon. The memory effect is caused by reversible structural changes suffered during modification. Buckled cell walls caused by compression are partially straightened during moisturizing. This process is limited in the case of large structural changes. Because there is a negligible difference in the lateral (and thus volumetric) dimension changes due to moisture changes in samples that were untreated, pleated, and fixated for a long time, it is necessary to deal with the dimensional changes in the fibre direction. In the shrinkage–swelling tests, the results are steady from the third drying–wetting cycle onwards. Both untreated and steamed samples have a shrinkage and swelling of 0.11%–0.16% in the longitudinal direction. The same is 0.8% for pleated beech and 1.3% for pleated oak (six to nine times greater), while the final values for samples fixated for a long time are 1.8%–2.1% and 2.4%–2.7% in cycles three and four, respectively (Figure 11). The increased shrinkage and swelling in the fibre direction is due to the deformation of the cell walls: the much higher lateral shrinkage partly appears in the longitudinal direction [111,128].

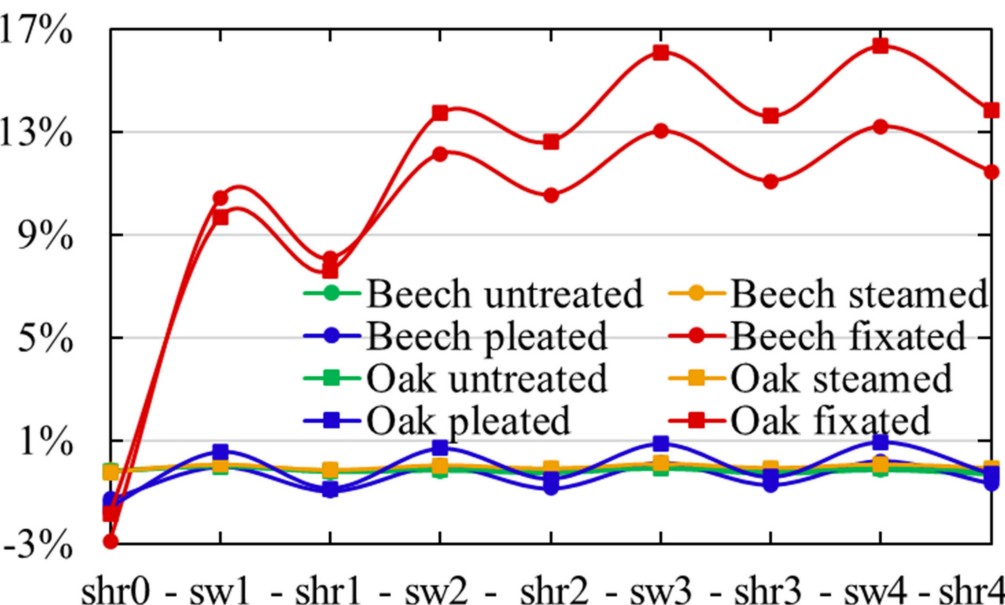

**Figure 11.** Cumulative plot of shrinkage–swelling values along the grain for the four cycles studied. Reproduced from [59], with permission from Báder, 2023. Abbreviations: shr—shrinkage cycle; sw—swelling cycle. Note: the shrinkage–swelling cycles are numbered; cycle 0 indicates the drying of the green wood samples as defined in the standard.

Compared to beech, the increase in length of the oak sample fixated for a long time is less pronounced in the first cycle, but its memory effect is more dominant in the second swelling cycle. The shrinkage results of cycles three and four shown in Figure 11 have been averaged by sample group and subtracted from the averages of the untreated values to obtain the memory effect. For pleated beech and oak samples, the memory effect is 0.19% and 0.14%, respectively, i.e., almost negligible, because the remaining shortening is low, about 3%–5%. Due to the high degree of spring back in the plastic state after pleating, less compression stress remains in the wood. For samples fixated for a long time, the memory effect is 11.6% and 13.9%, respectively. The remaining shortening after fixation for a long time is 12.5% and 18.4%, respectively. It can be concluded that plastic deformation is minimal for beech, but significant for oak. This implies that the 20% compression ratio in oaks is closer to their maximum compressibility than in beech. To easily calculate the swelling of pleated wood without the memory effect, see Equation (1); it is necessary to compare the saturated size of the specimen ($l_{rn}$) with its absolute dry size ($l_{r0+}$) in the next cycle, instead of the first absolute dry size ($l_{r0}$), as it is in the standard formula [107].

$$\alpha_{r+} = \frac{l_{rn} - l_{r0+}}{l_{r0+}} \cdot 100; \ [\%], \tag{1}$$

For oak, Báder and Németh [111] found that although its properties change below FSP, pleated wood shrinks after compression in its longitudinal direction in the moisture range even above FSP. The pleated sample shrank in the longitudinal direction by an average of 0.4% above the FSP, between 47% and 25% MC, followed by a shrinkage of 1.18% until 0% MC. The shrinkage over FSP may be because of the buckling of the cell walls, which may cause the formation of new available chemical links.

During the shrinkage–swelling tests, a weight loss (leaching) of the specimens occurs, denoted by Weight Percentage Loss (WPL). Pleating reduces WPL by 0.3% in beech and by 0.3% in oak. The beech and oak samples fixated for a long time show a WPL of 1.5% and 1.7%, respectively, i.e., WPL is 0.9% lower in beech and 0.4% lower in oak after treatment compared to untreated samples [107].

The pleating of wood produces only small differences in the hygroscopicity. Due to this process, the FSP is about 6% (MC%) lower at desorption compared to the native wood.

However, after adsorption, the FSP is the same for treated and untreated beech wood. Drying decreases the difference of the equilibrium moisture contents between treated and untreated samples, and below 10% MC, all MCs are the same. Pleating causes up to 1% (MC%) deviation of adsorption between untreated and treated samples. Under the same circumstances, the MC of pleated wood is never higher than the MC of untreated wood. This may be due to the partial degradation of hemicelluloses [86]. To investigate these phenomena, Báder et al. [129] carried out FTIR studies. This provides insight into the chemical changes in the thermo-hydro-mechanically modified beech and oak wood. Four stages of modification were investigated: untreated, steamed, compressed, and fixated for a long-time. As a result of THM treatments, the change of hydroxyl groups as well as the change of C–O and C–H functional groups of polysaccharides and lignin was observed. Using principal component analysis, the four sample groups of beech could be well separated, so beech is more sensitive than oak to the THM treatments under study. Using deconvolution and taking into account the slight change in hygroscopicity, it can be stated that although the amount of hydroxyl sites has not changed significantly, there are still significant differences in their location.

*5.3. Property-Influencing Factors during and after Pleating (Compression Rate, Fixation, Compression Ratio, and Moisture Content)*

During compression along the grain, the stress in specimens increases along with the compression rate. The change of MoR and bending stress is insignificant with the increasing compression rate. The MoE increases slightly, while the deflection at maximum load during four-point bending test decreases by 9% for oak and by 22% for beech, with increasing loading rates between 5 and 30%/min. However, the deflection still remains very high compared to untreated wood. Thus, Báder and Németh [130] preferred higher compression rates to achieve better productivity of the treatment.

The initial load of the compression is 7.5–12.5 MPa, which increases continuously up to 12.5–20.0 MPa until the specimen is shortened by 20% (Figure 12a).

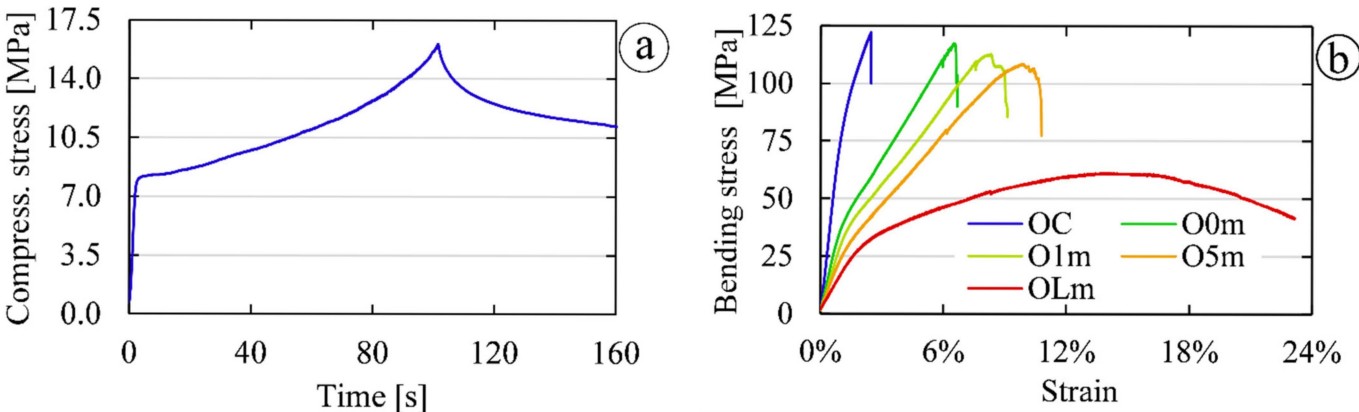

**Figure 12.** (**a**) Typical stress–time graph of compression followed by fixation for 1 min, reproduced from [110], licensed under <u>CC BY 3.0</u>; (**b**) Typical stress–strain graphs of oak samples for 4-point bending test data with different fixation times. Reproduced from [9], with permission from Pulp and Paper Research Institute, 2023. Abbreviations: OC—untreated sample; O0m, O1m, O3m, O5m—pleated samples with fixation for 0, 1, 3, and 5 min; OLm—pleated sample fixated for a long time.

The required compressive stress depends mostly on the wood species, the properties of the specimen, the compression ratio, and the compression rate [79]. If the compression process is followed by fixation, the compression stress continuously decreases, as shown in Figure 12a. Báder and Németh [9] showed that the compression stress reduction during fixation is in good correspondence with the remaining shortening of the material. Both are indicators of the change of the MoE and the flexibility of the material after the treatment.

According to Báder and Németh [9], in four-point bending tests, the force required for the same deflection is halved by pleating.

The results from the literature are somewhat uncertain because not all data are available, such as fixation time, which has a strong impact on mechanical results based on work by Báder and Németh [9]. For example, Vorreiter [102] gives a fixation time of 10–15 min after compression based on earlier experiences. Increasing the compression ratio and increasing the fixation time both decrease the MoE, similarly to the increase of the MC before bending [112]. Using a 20% compression ratio, with increasing fixation time the bendability of wood increases and the required bending force decreases proportionally for the same bending radii [9] (Figure 12b). The peaky graphs of the untreated specimens gradually become rounded both by compression and by increasing fixation time (Figure 12b). Finally, specimens fixated for a long time become ductile and can undergo significant plastic deformation before fracturing [9]. A very similar effect can be observed in the dissertation of Kuzsella [74] as a result of the increasing compression ratio (Figure 9). Pleated wood and wood that has been fixated for a long time have very different properties, as do untreated and pleated wood. As a general trend, a longer fixation time increases the effects of the compression along the grain [119]. Báder [107] compared different fixation times. Analysis of the decrease of compression stress during fixation suggests that at least the last 10 h of the 18 h fixation time are unnecessary. Therefore, oak and beech wood species were fixated for 18, 5, and 3 h, showing that fixation for 3 h results in very similar changes to fixation for 18 h in terms of MoR, MoE, bending stress, deflection at maximum load, and maximum deflection. In other words, for fixation for a long time, it is sufficient to keep the specimen in a compressed state for 3 h.

As the ratio of compression increases, the MoE decreases, i.e., the material can undergo greater deformation prior to failure and is more easily shaped [73]. With increasing compression ratio, the decreasing slope of the linear section of the graphs represents the reduction of the MoE (Figure 9). Figure 13 shows the average changes of the MoR, MoE, strain at the moment of the greatest bending stress, and specific fracture energy as a result of different compression ratios.

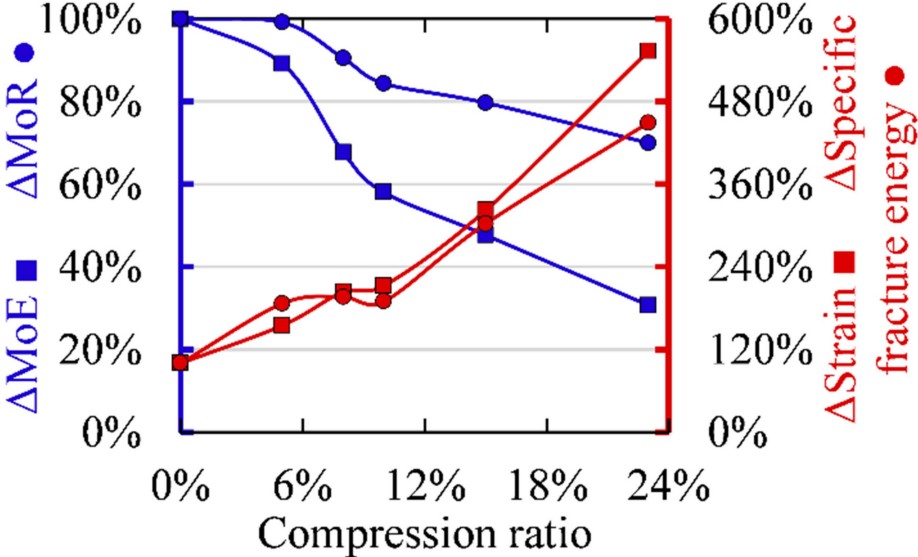

**Figure 13.** Relative results of a 3-point bend test as a function of beech species with different compression ratios. Numerical test results from [74] were used to plot the graph, reproduced from [59] with permission from Báder, 2023. Abbreviations: ΔStrain—change of the strain at the moment of the greatest bending stress; ΔMoR—change of the modulus of rupture; ΔMoE—change of the modulus of elasticity.

MC highly affects the mechanical properties of wood. Báder and Németh [112] determined that the change ratio of untreated and treated wood differs in compressive

strength parallel to the grain in the $k_{bend}$ and in the MoE. The coefficient of moisture dependence of a mechanical property ($\alpha$) is 3.2% for compressive strength parallel to the grain. The value of $\alpha$ is 4.2% both for the MoR and $k_{bend}$, 5.0% for MoE, 5.4% for bending stress, and 7.6% for the highest deflection during a four-point bending test. MC must be close to the FSP for the best pliability while bending the pleated wood.

### 5.4. Changes at the Cellular Level: The Relationship between Buckling and Bending Elongation

The stresses generated during bending can be divided into two main parts: compressive stress on the concave side and tensile stress on the convex side [64,102,131,132]. The boundary between these two is the neutral axis [102]. On the tensed side, wood exerts a higher resistance, but allows much less deformation compared to the compressed side, such that the fracture during bending usually occurs on the side under tension [102,132,133] (Figure 14).

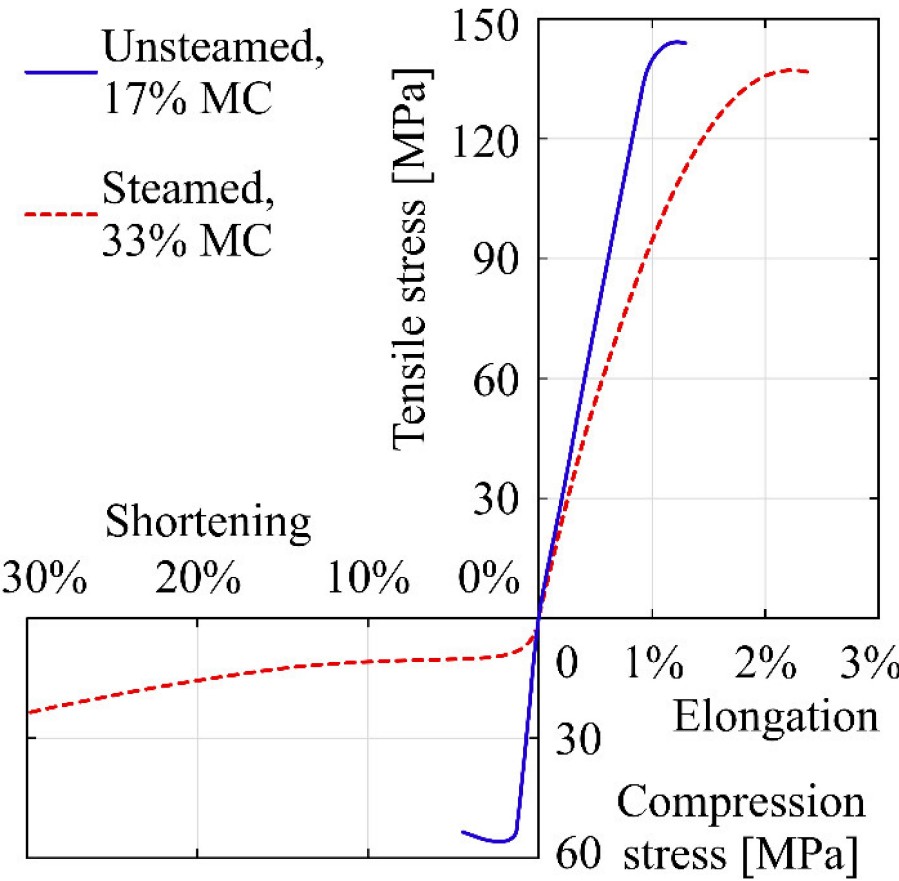

**Figure 14.** Stress–strain graph of beech wood. Reproduced from [59], with permission from Báder, 2023.

To avoid fissures and fractures while bending wood, the neutral thread should be brought as close as possible to the tensed side using a steel restraining band, which does not allow the elongation of wood during bending [102,131]. If the entire cross-section is free of tensile stress, the wood will not be damaged during bending because of the tensile stresses.

As a result of pleating, the structure of wood modifies, which causes a change in its mechanical and other physical properties [77]. The fibres and other cellular constituents ensure the longitudinal stiffness of the wood. Their position and shape change with the compression, but their structure remains unchanged [18,77]. According to the descriptions of Deibl et al. [12], Compwood Machines Ltd. [75], and Kovács et al. [69], after plasticization, wood becomes compressible, and the process causes the buckling of the cell walls of wood. Studies of microscopic changes have shown that 98% of the tracheas of beech wood

are already buckled when the 15% compression ratio is reached [74]. Scanning electron microscopy (SEM) images show changes in wood structure due to pleating (Figure 15).

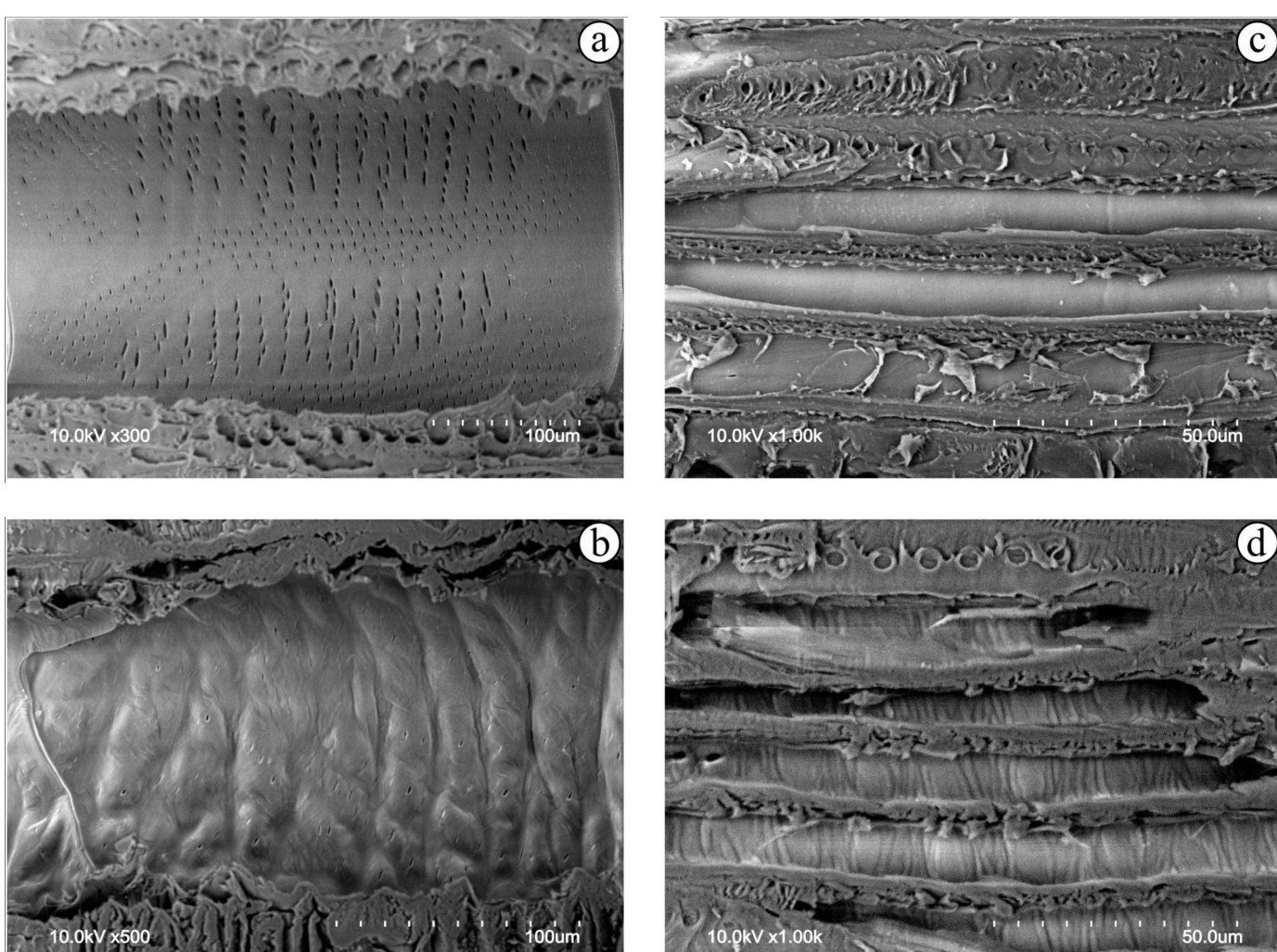

**Figure 15.** Scanning electron microscopy images of oak wood. (**a**) Untreated trachea (10 kV ×300); (**b**) pleated trachea (10 kV ×500); (**c**) untreated fibres (10 kV ×1000); (**d**) pleated fibres (10 kV ×1000). Reproduced from [9], with permission from Pulp and Paper Research Institute, 2023.

According to many authors, the changes of the cell walls resemble a half-closed concertina, or a plisse shade [3,9,14,62,74,133]. Due to the changes in the cell walls, the compression process along the grain was described as "crushing" by Navi and Pizzi [91], but the wood fibres and tracheas do not really fracture as the word "crushing" implies. The structure and properties will be changed according to the expectations of the modification process and the wood remains useable, contrary to the term "crushed." A more precise and descriptive term for the phenomenon is "pleating" [9]. Pleating should not be confused with densification, which means the compressing of softened wood in the transverse direction at high temperature, patented by Stöckhart in 1886 [91]. While the purpose of densification is to improve hardness and other mechanical properties by increasing the density of wood, the goal of pleating is to improve its pliability.

During bending, the buckled cell walls can straighten on the tensed side; thus, pleated wood has a reserve compared to untreated wood, even during much greater deformation [88]. This fact appears in the patent of Hanemann [133], and Sadadt nezhad et al. [120] also show that by using a lower tensile stress, a significantly higher elongation can be achieved compared to untreated wood. The results obtained by Vorreiter [18] regarding

the higher force required to split pleated wood can be explained by the buckling of the longitudinal cells and the alteration of their position. It is more difficult to separate fibres that run both parallel and also crosswise (Figure 8).

It is important to keep in mind that the most significant mechanical element of the hardwoods is the wood fibre. It has a thicker cell wall than trachea and its main function is reinforcement [77,134,135]. Based on the SEM analysis by Báder and Németh [9], the wall of the wood fibres is less prone to buckling, but due to a high-ratio compression and fixation for a long time, these changes appear similarly to the trachea. If the specimen is unable to bend or break or it does not have any other mechanical damage as a result of pleating, the microscopic changes must necessarily occur in all wood species that can be uniformly compressed along their length. Thus, compared to its untreated state, wood will be only slightly elongated or will not be elongated at all during bending as a result of pleating. As such, it can be bent into a much smaller radius before it fractures.

The cell wall changes but is not destroyed, thus ensuring the constant pliability of wood [11,18]. Ivánovics [77] concluded that the bundles of fibres move in the plasticized matrix and inter-cellular cracks may also appear. The cells may become longitudinal movements along adjacent cell walls and cell walls thicken. Both changes result in an increase in the cross-section of the cells and the material also penetrates the lumens. A similar conclusion was reached earlier by Hanemann [133] and by Stevens and Turner [62] that many slip planes or initial compression failures can be observed microscopically but no fracture, split, or disintegration of the material as a whole occurs. The SEM images of Sadadt nezhad et al. [120] show the splitting of the cell walls (Figure 16). Compression generates cracks in the cell walls and in the intercellular spaces. Their continuity is interrupted, which weakens the binding and resistance of the cells to applied forces, creating a flexible and unstable state. The microscopic cracks extend in the course of the drying process, allowing the cells to move more easily and freely. The changes observed during the tensile tests (significantly higher elongation at lower tensile stress) prove the breaking of the cellulose chains and the weakening of the matrix [120].

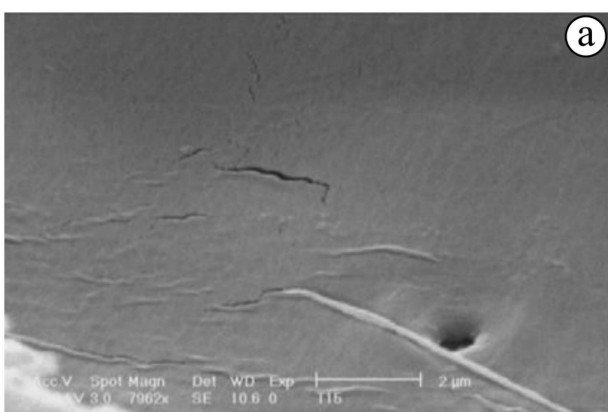 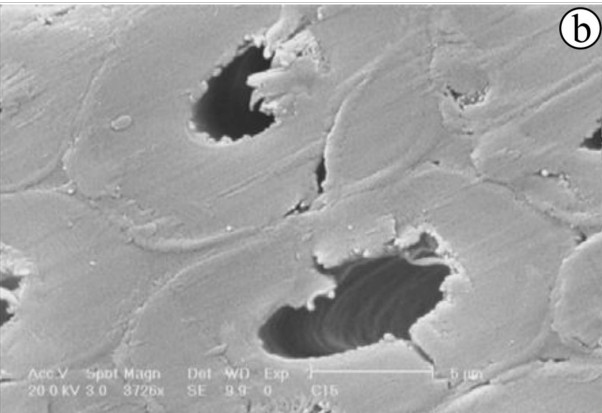

**Figure 16.** Scanning electron microscopy images of (**a**) the inner portion of the cell wall (20 kV ×7962) and (**b**) cross-section of the specimen (20 kV ×3726) as a result of 15% compression ratio of beech wood. Adapted from [120], licensed under CC BY 4.0.

The changes occurring under the influence of a high compression ratio are failures at the cellular level, which become visible to the naked eye. These start as microscopic cracks and may become longitudinal movements [77]. Kánnár [136] observed similar fractures in compression tests: longitudinal cells bend over a short section, possibly having shear deformations. As the load increases, these deformations extend to the entire cross-section, which is accompanied by the destruction of the cell wall (splits, cracks, and separation of layers). Further increasing the deformation destroys more cross-sections while the longitudinal stress values remain constant. Finally, the visible separation begins at the borderline between fibres and rays.

The micromechanical characterization of pleated wood was investigated by nanoindentation (NI) and atomic force microscopy (AFM) by Báder et al. [119]. For AFM investigations, smooth surfaces were cut parallel to the fibre direction. With the treatment, AFM images showed microfibril disorientation in the S2 cell wall layer (Figure 17). Although only 2D orientation is visible in Figure 17, it is expected that crinkling arises in 3D; thus, the microfibrils are often intersected.

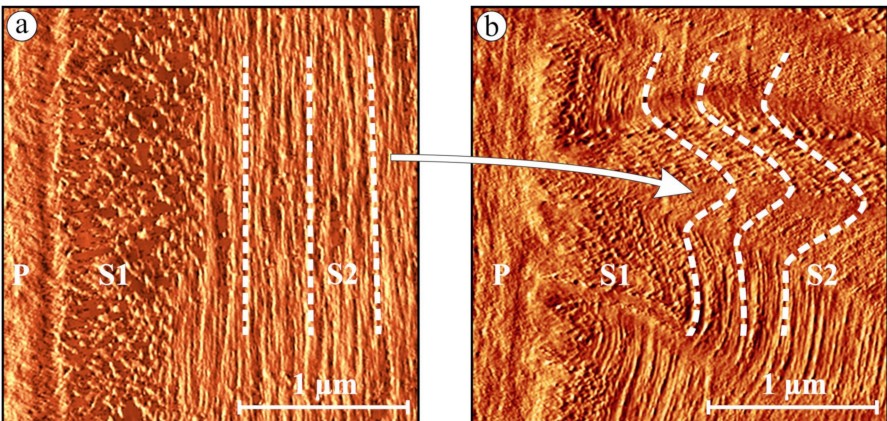

**Figure 17.** (**a**) Atomic force microscopy images on radial sections of an oak wood cell wall in an untreated specimen and (**b**) in a compressed specimen exposed to a fixation for a long time. The images show the primary cell wall (P), the layers of the secondary cell wall (S1 and S2), and the corresponding microfibril direction in the S2 layer with dotted lines. The arrow symbolises the compression process, what the cell wall looked like before and what it looked like after. Reproduced from [119], licensed under CC BY 4.0.

Oak and beech samples showed similar micromechanical changes and trends in their NI as a result of the treatment. The indentation modulus of the secondary cell wall S2 layer was investigated parallel to the grain. It decreased by 48% due to 20% compression along the grain and a fixation for a long time, but the hardness of the S2 layer was only slightly affected. The total deformation work increased by 14% for the treated wood compared to untreated wood; the viscoelastic part of the total deformation work remained unchanged, while the elastic part of the total deformation work increased by 9% at the expense of plastic deformation work. Thus, the treatments resulted in higher resilience and reduced ductility of the cell walls compared to the untreated wood. This finding seems to contradict the macromechanical results, where the decreased MoE results in high ductility of the sample. The decrease in the MoE and compression stress by the treatment is explained by the systematically deformed fibres. On the ultrastuctural level, the disoriented microfibrils result in a reduced indentation modulus value. According to the reviews of Báder et al. [119], the macromechanical changes, which result from the buckling of the cell walls (Figure 15), outweigh the changes in the microfibrils (Figure 17) that would cause the opposite micromechanical behavior. Thus, the crinkles and/or cracks of the microfibrils and the buckling of the cell walls explain the changes both in the indentation modulus and the MoE.

### 5.5. Further Treatment Possibilities and Their Effects (Surface Treatment, Impregnation, Heat Treatment, and Fungal Resistance)

Szántó [137] investigated the adhesion of water-based and organic solvent-based varnishes on both tangential and radial anatomical surfaces of untreated and pleated beech and oak samples. Pleating has not affected or increased the adhesion of varnishes, except for the solvent-based varnish on tangential beech surfaces; there was an 11.0% decrease. The highest increase in adhesion was measured for solvent-based varnish on tangential oak surfaces (10.3%). Samples treated with aqueous varnish also showed the greatest increase in adhesion on the tangential side of beech (7.5%). Interestingly, these results are not fully

consistent with the results of the goniometric tests on tangential surfaces: the contact angle was not changed by the pleating of oak but increased in beech.

Samples of beech and oak heartwood that were untreated, steamed, pleated, and fixated for 1 min were impregnated with lactic acid (LA) by Báder and Németh [128]. The longitudinal shrinkage and swelling increased by 580% and 660% for beech and by 790% and 900% for oak, caused by pleating. Before impregnation, the LA monomer was dehydrated followed by oligomerisation. After LA treatment, the dimensional changes in the lateral directions halved. In the longitudinal direction, the dimension changes became close to the shrinkage and swelling of untreated specimens. Oak could not be impregnated with LA. LA treatment is an effective solution to greatly decrease the dimensional changes of pleated beech wood and other wood species suitable for impregnation. The LA-treated wood takes on a pleasant brownish colour. This wood–plastic composite system is environmentally friendly, containing only biodegradable components [138].

The 180 °C heat treatment by Grászli [122] made pleated wood more brittle and, therefore, the compressive strength of beech increased by 14% and that of oak increased by 31%, while their compressive modulus of elasticity doubled. Thus, the compressive test results of the pleated and heat-treated samples are better than those of the untreated samples.

In terms of lateral Brinell hardness, Grászli [122] found a 28% increase for beech and 16% increase for oak, compared to pleated samples. The increase in end-grain Brinell hardness was 18% and 28%, respectively. The increased brittleness had a good effect on the hardness values.

The heat treatment of pleated beech and oak wood was also investigated by Horváth [126]. The treatment was carried out at 160 °C and 200 °C for 10 h under atmospheric conditions. For oak sample fixated for a long time, the significant memory effect (10.5%) over two saturation–drying cycles is halved by heat treatment at 160 °C, while heat treatment at 200 °C eliminates it. The same trend is observed for beech, but with a slightly lower memory effect. This is also true for pleated samples, but of course to a lesser extent; 1.3% is the maximum increase in length due to the memory effect, which is reduced by the heat treatments.

The heat treatment at 160 °C after the pleating methods causes only smaller changes, while the heat treatment at 200 °C almost halves the lateral shrinkage and swelling of the tested wood species. Unfortunately, Horváth [126] did not separate the shrinkage and swelling parallel to the grain from the length change caused by the memory effect, so only the second shrinkage–swelling cycle is analysed here. Curiously, after heat treatment at 160 °C, the swelling and shrinkage values of the beech sample fixated for a long time increased by 30% and 100%, respectively, while there was no significant difference elsewhere. However, in all cases, heat treatment at 200 °C reduced shrinkage and swelling, typically by a half to a third. The equilibrium moisture content at 20 °C and 65% relative humidity varied uniformly regardless of the pleating method and was only affected by heat treatment: 11%–13% for untreated wood species, 7%–9% for wood species heat-treated at 160 °C, and 4%–6% for wood species heat-treated at 200 °C for both wood species.

Mechanically, the sample fixated for a long time and then heat-treated at 160 °C became more brittle, while all other samples became extremely brittle, showing similar or even lower deformation ability in bending tests compared to the uncompressed samples. The same trend was observed for MoR, while MoE was hardly affected by heat treatment. The average MoR varied between 130 and 80 MPa for uncompressed oak and between 120 and 30 MPa for pleated oak, depending on the intensity of the heat treatment. Thus, heat treatment increases the compression-induced differences. The MoR of the specimens fixated for a long time increases from 60 to 70 MPa to above 80 MPa after heat treatment at 160 °C, but decreases to 15–20 MPa after heat treatment at 200 °C. The same trend is observed for beech [126]. Consequently, heat treatment at 200 °C cannot be used effectively after pleating.

Horváth et al. [139] investigated the resistance to white rot fungi (*Trametes versicolor* (L.) Lloyd) of differently treated beech wood. Pleated specimens showed a mass loss similar

to their untreated counterparts. Fixation for a long time reduced the resistance to fungal decay. Samples pleated and heat-treated (fungal decay ratio: 28.84%) and pleated and impregnated with LA (fungal decay ratio: 18.31%) showed improved resistance compared to the untreated ones (fungal decay ratio: 31.62%). Specimens impregnated with lactic acid bind a high amount of moisture, which may cause problems with dimensional stability. According to Horváth [140], the mass losses of oak samples were much smaller after the fungal decay tests with *Trametes versicolor* (L.) Lloyd. In addition to the 16.24% average weight loss of the untreated sample, the pleated sample suffered a weight loss of 11.26% and the sample fixated for a long time had a mass loss of 12.30%. The sample pleated and heat-treated had a mass loss of 9.32%.

Horváth [140] examined beech samples exposed to brown rot fungus (*Antrodia sinuosa* (Fr.) P. Karst). Compared to the fungal decay of the untreated sample (averagely 20.90%), all treatments gave similar or slightly poorer results. No remarkable differences were observed. Although mineralized (wood was impregnated with an aqueous solution of potassium oxalate and then calcium chloride) samples showed low weight loss for both fungal species, their test could not be evaluated due to their extremely high moisture content. This may be due to the potassium chloride remaining after the mineralization treatment.

As can be seen in Section 5, a wide range of test results are available for pleated wood produced by different methods. However, as in all disciplines, there are and will always be gaps in knowledge of pleated wood. At the moment, it seems that the most important thing would be to obtain the tensile test parameters and to prove or disprove the theory of the shift of cells relative to each other. Catalysts that enhance the softening of cell walls could be sought to improve yield and increase productivity. Today, it is not yet possible to say with certainty whether or not a timber will be successfully compressible. A solution could be the prior sorting and selection of the timber to be treated using non-destructive test methods. It would be useful to gain knowledge about the applicability of tropical wood species and mostly tropical plantation wood species from sustainable farming (e.g., teak, some eucalypt species, bangkirai, jatoba, etc.) and to carry out life cycle assessment (LCA) analyses to see if pleated wood is indeed a "green alternative" to non-renewable materials. Of course, the application of new testing technologies, as technology develops, may bring even more new results and knowledge, while at the same time raising new questions.

## 6. Conclusions

At the beginning of the twentieth century, a wood-bending method was developed with the aim of simplifying and improving wood-bending technology. The compression of wood along its grain (pleating) seemed to be a viable opportunity in this field. Using this treatment, wood can be bent into a smaller radius and it is less likely to fracture from multidirectional bending and twisting compared to the classic plasticized wood. The first patent was granted in 1917, which contains the basic principles of the process. Nowadays, industrial technology is available for serial production, utilizing the achievements of the modern age. This study fills the need for a review of pleating, which has been missing over the last 100 years.

For pleating and bending, high-quality, clear hardwood species are required with a uniform structure and a higher density (oak, beech, ash, maple, cherry, walnut, etc.). After plasticizing wood with a minimum initial moisture content of 20%, the compression device prevents the deflection caused by the pressure in the longitudinal direction. Wood can be compressed by 10%–30% of its original length depending on the wood species and the requirements of the final product. Generally, 20% compression ratio is used. Eliminating the pressure after compression, wood springs back and has a remaining shortening of 3%–5% of its initial length, and it will be bendable as long as the moisture content stays above 16%. If wood is cooled down in its compressed state (fixation), the spring back will be much less and the effects of pleating will be much stronger. After compression, the wood is immediately suitable for bending, but it can also be stored under appropriate conditions if necessary. In order to keep the workpiece permanently in a desired shape, conventional

tools can be used for bending, and after a complete drying, wood retains its set shape for an arbitrary time and can be machined as a normal piece of wood. Areas of application include furniture manufacturing, interior design, sports and musical instrument manufacturing, modeling, fine arts, restoration, etc. This product is free of chemical additives and is environmentally friendly.

Pleating reduces the modulus of elasticity to one-third and the modulus of rupture by one-third for oak and beech; furthermore, bending stress is halved. The energy absorbed until the fracture multiplies, so compressed wood exhibits a more viscoelastic behavior than untreated wood. The impact bending strength triples; the specific impact work also increases, i.e., pleated wood is more ductile than untreated wood. The procedure provides a bending ratio (thickness/smallest bending radius) above $\frac{1}{2}$. Almost all the cell walls buckle until the 15% compression ratio is reached, which is why this process is called "pleating". During a well-performed compression, the cells may become longitudinal movements relative to each other and the cell walls thicken and buckle, thereby providing an elongation reserve for bending. The microfibrils of the S2 cell wall layer, which provides the rigidity of the cell, crinkle, thereby significantly reducing the indentation modulus of the cell wall. Materials used for surface treatment adhere well to the surface of pleated wood, and various processes (heat treatment and lactic acid treatment) can be used to improve its dimensional stability, fungal resistance, and other properties in an environmentally friendly way. Further improvements in pleating and research on the properties of pleated wood are needed, e.g., exploration of tensile properties, simplification of the pleating process, applicability to tropical plantation wood species, etc. As a result of ongoing tests at the University of Sopron, Hungary, scientific knowledge of pleated wood constantly increases, and further results are expected in the future.

**Author Contributions:** Conceptualization, M.B.; Methodology, M.B.; Validation, M.B. and R.N.; Formal Analysis, M.B.; Investigation, M.B.; Resources, M.B. and R.N.; Writing—Original Draft Preparation, M.B.; Writing—Review and Editing, M.B. and R.N.; Visualization, M.B.; Supervision, R.N.; Project Administration, R.N.; Funding Acquisition, R.N. All authors have read and agreed to the published version of the manuscript.

**Funding:** The publication was made in frame of the project TKP2021-NKTA-43, which has been implemented with the support provided by the Ministry of Culture and Innovation of Hungary from the National Research, Development and Innovation Fund, financed under the TKP2021-NKTA funding scheme.

**Data Availability Statement:** Not applicable.

**Acknowledgments:** The authors would like to thank Gergely Ivánovics for his useful insights and help in data collection. Compwood Products Ltd. is acknowledged for demonstrating the industrial pleating process.

**Conflicts of Interest:** The authors declare no conflict of interest.

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
