# Peer review of "A Review of Wood Compression along the Grain—After the 100th Anniversary of Pleating"

_forests, doi:10.3390/f14040763_

Round 1

Reviewer 1 Report

The aim of the paper is to review the history of the process of wood compression along the grain. Its main contribution is to summarize in one place the different production methods and the properties of compressed wood.

Overall, the review paper is well-written, it contains a comprehensive list of sources and the topic is of relevance to the field. However, the gap in knowledge should be clearly identified and lightened. Further research directions should be addressed both in the main text and the Conclusions chapter.

Title

Maybe "A Detailed Review of" phrase should be omitted.

Introduction

The part from the second paragraph should be renamed to 2. History of pleating. This part should be shortened, while the first paragraph should be extended and should contain the basic information related to the pleating process (definition, the basic explanation of the process itself...).

Please delete the word Methods in the title of chapter 3.

Fig. 7 should be simplified in a way that represents 3 main steps (plasticization, compression, posttreatment) and moved to the beginning of the chapter.

Please delete the word Results in the title of chapter 4.

Conclusions

The gap in knowledge should be identified and further research directions recommended.

Reviewer 2 Report

The text has been written comprehensively, from the history of pleating wood to its application in laboratory and industrial scales. The technical aspects related to the treatment process, including pre-treatment, post-treatment, and the mechanisms that occur in the wood as a result of the treatment, up to the ultrastructure level, are also presented in the text. Physical and mechanical characteristics are informed based on the type of wood sample. However, one aspect that could be improved is the inclusion of information on the application of the treatment and pleating products on non-subtropic wood types. This would provide a useful comparison to the development of compression of wood along the grain or pleating wood, given the long history of this type of treatment. Such information would further enhance the understanding of the potential application of pleating wood beyond subtropic wood types.
